# Differentially Private Synthetic Data via APIs 3: Using Simulators Instead of Foundation Models

## Abstract

Differentially private (DP) synthetic data, which closely resembles the original private data while maintaining strong privacy guarantees, has become a key tool for unlocking the value of private data without compromising privacy. Recently, Private Evolution (PE) has emerged as a promising method for generating DP synthetic data. Unlike other training-based approaches, PE only requires access to inference APIs from foundation models, enabling it to harness the power of state-of-the-art (SoTA) models. However, a suitable foundation model for a specific private data domain is not always available. In this paper, we discover that the PE framework is sufficiently general to allow APIs beyond foundation models. In particular, we demonstrate that many SoTA *data synthesizers that do not rely on neural networks*—such as computer graphics-based image generators, which we refer to as *simulators*—can be effectively integrated into PE. This insight significantly broadens PE's applicability and unlocks the potential of powerful simulators for DP data synthesis. We explore this approach, named Sim-PE, for image synthesis. Across four diverse simulators, Sim-PE performs well, improving the downstream classification accuracy of PE by up to $3\times$, reducing FID by up to 80%, and offering much greater efficiency. We also show that simulators and foundation models can be easily leveraged together within Sim-PE to achieve further improvements.

## 1 Introduction

Leaking sensitive user information is a major concern in data-driven applications. A common solution is to generate differentially private (DP) (Dwork et al., 2006) synthetic data that resembles the original while ensuring strong privacy guarantees. Such data can substitute the original in tasks like model fine-tuning, statistical analysis, and data sharing, while preserving user privacy (Bowen & Snoke, 2019; Lin, 2022; Tao et al., 2021; Hu et al., 2024).

Private Evolution (PE) (Lin et al., 2023; Xie et al., 2024) has recently emerged as a promising method for DP data synthesis. It begins by probing a foundation model to produce random samples, then iteratively selects those most similar to private data and uses the model to generate more like them. Unlike prior state-of-the-art (SoTA) methods that fine-tune open-source models, PE relies solely on model inference–making it up to $66\times$ faster (Xie et al., 2024). More importantly, this allows PE to easily leverage cutting-edge foundation models like GPT-4 (OpenAI, 2023) and Stable Diffusion (Rombach et al., 2022), achieving SoTA performance on multiple image and text benchmarks (Lin et al., 2023; Xie et al., 2024; Hou et al., 2024; Zou et al., 2025; Hou et al., 2025; Wang et al., 2025a;b). PE has also been adopted in Microsoft and Apple (Apple, 2025; Afonja et al., 2024).

However, PE relies on foundation models suited to the private data domain, which may not always be available. When the model's distribution significantly differs from the private data, PE's performance lags far behind training-based methods (Gong et al., 2025).

To address this question, we note that in the traditional synthetic data field—where private data is *not* involved—*non-neural-network data synthesizers* remain widely used, especially in domains where foundation models struggle. Examples include computer graphics-based renders for images, videos, and 3D data (e.g., Blender (Community, 2018) and Unreal (Epic Games)), physics-based simulators for robotics data (e.g., Genesis (Authors, 2024)), and network simulators for networking data (e.g., ns (Issariyakul et al., 2009; Riley & Henderson, 2010)). For brevity, we refer to these tools as *simulators*. While these simulators have been successful, their applications in *DP data synthesis*

remain underexplored. This is understandable, as adapting these simulators to fit private data in a DP fashion requires non-trivial, case-by-case modifications. Our key insight is that PE only requires two APIs: RANDOM_API that generates random samples and VARIATION_API that generates samples similar to the given one. These APIs do not have to come from foundation models. Thus, we ask: *Can PE use simulators in place of foundation models?* If viable, this approach could greatly expand PE's capabilities and unlock the potential of a wide range of simulators for DP data synthesis.

In this paper, we propose Sim-PE (Fig. 1) to exploit this potential for *image* generation. We consider two types of simulator access: **(1) The simulator is accessible.** We define RANDOM_API as rendering an image with random simulator parameters, and VARIATION_API as slightly perturbing the simulator parameters of the given image. **(2) The simulator is inaccessible—only its generated data is released.** This scenario is quite common (Wood et al., 2021; Bae et al., 2023), especially when simulator assets are proprietary (Kar et al., 2019; Devaranjan et al., 2020). In this case, we define RANDOM_API as randomly selecting an image from the dataset, and VARIATION_API as randomly selecting a nearest neighbor of the given image. We demonstrate that with suitable simulators, Sim-PE can outperform PE with foundation models. Our key contributions are:

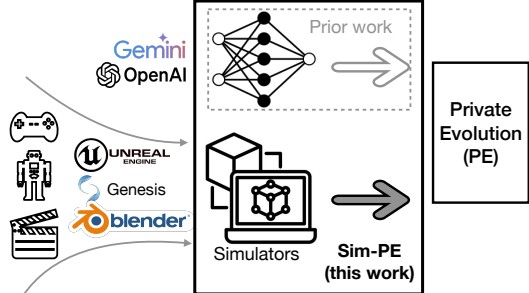

Figure 1: Unlike prior PE work that relies solely on foundation models, we show that PE is also compatible with non-neural-network data synthesis tools, which we call *simulators*. This greatly broadens PE's applicability and enables SoTA simulators for DP data synthesis.

- **Advancing PE.** We discover that PE can leverage tools beyond foundation models and propose Sim-PE—an extension that uses simulators, significantly broadening PE's applicability. We also introduce the use of *both foundation models and simulators interchangeably* during synthesis, allowing for the benefits of both to be leveraged through PE's easy and standardized interface.
- **Bringing simulators to DP synthetic data.** Although simulators are widely used and powerful (App. I.1), they have been largely absent from DP data generation. Sim-PE is the first framework to unlock their potential in this space.
- **Results.** We demonstrate promising results with Sim-PE. For instance, on the MNIST dataset with $\epsilon = 1$, downstream classification accuracy increases to 89.1%, compared to 27.9% with the original PE. Furthermore, combining foundation models with weak simulators results in improved performance compared to using either one alone.

## 2 PRELIMINARIES AND MOTIVATION

### 2.1 PRELIMINARIES

**Synthetic data** refers to "fake" data generated by models or software for various applications, including data augmentation, model training, and software testing (Lin, 2022). While neural-network-based generative models such as GANs (Goodfellow et al., 2020), diffusion models (Sohl-Dickstein et al., 2015), and auto-regressive models (OpenAI, 2023; Liu et al., 2024a) are widely used, non-neural-network tools remain SoTA in many applications. For example, ns (Issariyakul et al., 2009; Riley & Henderson, 2010) can simulate networks and generate network packets based on network configurations. Blender (Community, 2018), given 3D models and lighting configurations, can render images and videos, and is extensively used in movie production. **In this paper, we refer to these tools as *simulators*.** See App. I.1 for a discussion on the continued importance of simulators, even in the era of large foundation models.

**DP synthetic data** requires the synthetic data to be close to a given private dataset, while having a strict Differential Privacy (DP) (Dwork et al., 2006) guarantee. Formally, a mechanism $\mathcal{M}$ is $(\epsilon, \delta)$-DP if for any two neighboring datasets $\mathcal{D}$ and $\mathcal{D}'$ (i.e., $\mathcal{D}'$ has one extra entry compared to $\mathcal{D}$ or vice versa) and for any set $S$ of outputs of $\mathcal{M}$, we have $\mathbb{P}(\mathcal{M}(\mathcal{D}) \in S) \leq e^{\epsilon}\mathbb{P}(\mathcal{M}(\mathcal{D}') \in S) + \delta$. Smaller $\epsilon$ and $\delta$ imply stronger privacy guarantees. Current SoTA DP image and text synthesis methods typically requires neural network training (Lin et al., 2020a; Beaulieu-Jones et al., 2019; Dockhorn et al., 2022; Yin et al., 2022; Yu et al., 2021; He et al., 2022; Li et al., 2021; Ghalebikesabi et al., 2023a; Yue et al., 2022; Jordon et al., 2019; Harder et al., 2023; 2021b; Vinaroz et al., 2022; Cao et al., 2021; Chen et al., 2022).

**Private Evolution** (PE) (Lin et al., 2023; Xie et al., 2024) is a recent training-free framework for DP data synthesis. PE only requires *inference access* to the foundation models. Therefore, unlike prior training-based methods, PE can leverage the SoTA models even if they are behind APIs (e.g., GPT-4) and is more computationally efficient (Lin et al., 2023; Xie et al., 2024). In more detail, PE has achieved SoTA performance on several *image* and *text* benchmarks (Lin et al., 2023; Xie et al., 2024; Wang et al., 2025a;b). *When using similar open-source pre-trained models*, PE attains an *image* quality score of FID $\leq 7.9$ on CIFAR10 with a privacy cost of $\epsilon = 0.67$, a significant improvement over the previous SoTA, which required $\epsilon = 32$ (Lin et al., 2023). Furthermore, PE can be up to $66\times$ more efficient than training-based methods on DP *text* generation (Xie et al., 2024). *By leveraging SoTA models behind APIs—where training-based methods are not applicable—*PE further enhances performance, outperforming all prior approaches in downstream *text* classification accuracy on the OpenReview dataset (Xie et al., 2024). Additionally, PE can be applied in *federated learning* to shift model training from devices to central servers in a differentially private and more efficient manner (Hou et al., 2024; Zou et al., 2025; Hou et al., 2025). Moreover, PE has been adopted by some of the largest IT companies such as Microsoft and Apple (Apple, 2025; Afonja et al., 2024).

## 2.2 MOTIVATION

While PE achieves SoTA performance on several image and text benchmarks (Lin et al., 2023; Xie et al., 2024; Hou et al., 2024), its performance significantly drops when there is a large distribution shift between the private data and the foundation model's pre-trained data (Gong et al., 2025). For instance, when using the MNIST dataset (LeCun, 1998) (handwritten digits) as the private data, training a downstream digit classifier (10 classes) on DP synthetic data (with $\epsilon = 1$) from PE—using a foundation model pre-trained on ImageNet—yields an accuracy of only 27.9%. Since relevant foundation models may not always be available for every domain, this limitation hinders PE's applicability in real-world scenarios. Extending PE to leverage simulators could significantly expand its potential applications.

More broadly, as discussed in § 2.1 and App. I.1, simulators cannot be substituted by foundation models in (non-DP) data synthesis across many domains. Unfortunately, current SoTA DP synthetic data methods are deeply reliant on machine learning models (e.g., requiring model training) and cannot be applied to simulators. By extending PE to work with simulators, we aim to unlock the potential of simulators in DP data synthesis.

## 3 SIM-PE: PRIVATE EVOLUTION (PE) WITH SIMULATORS

In this paper, we focus on DP *image* generation. A key advantage of the PE framework is that it decouples the DP mechanism from the data generation backend. Specifically, any backend that supports (1) RANDOM_API, which generates a random sample (e.g., a random bird image), and (2) VARIATION_API, which produces slight variations of a given sample (e.g., a similar bird image), can be integrated into PE and turned into a DP data synthesis algorithm. Prior work on PE (Lin et al., 2023; Xie et al., 2024; Hou et al., 2024; Zou et al., 2025; Hou et al., 2025; Swanberg et al., 2025; Wang et al., 2025a;b) has exclusively used foundation models to implement these APIs. Our key insight is that these APIs do not need to be powered by foundation models: traditional data synthesizers that do not rely on neural networks—referred to as *simulators*—can also be used to implement RANDOM_API and VARIATION_API.

In the following sections, we first provide an overview of the Sim-PE algorithm (§ 3.1) and the design of its APIs (§ 3.2 and 3.3), then discuss how simulators and foundation models can be jointly used in Sim-PE to leverage the strengths of both (§ 3.4).

### 3.1 OVERVIEW

**Algorithm overview.** Except for the APIs, Sim-PE largely follows the same workflow as PE. For completeness, we briefly describe the workflow here and include the full algorithm in App. A. We first use RANDOM_API to generate an initial set of random samples (Line 4). Then, we iteratively refine these samples using the private data. In each iteration:

- Each private sample casts a vote for its closest synthetic sample. This yields a histogram (denoted DP_NN_HISTOGRAM) reflecting how well each synthetic sample aligns with the private data (Line 6). To ensure differential privacy, Gaussian noise is added to this histogram.
- We sample synthetic data according to the noisy histogram, giving higher likelihood to those samples that align more closely with private data (Lines 7 and 8).

- We apply VARIATION_API to the drawn samples to generate additional variants (Line 9). These samples become the initialization of the next iteration.

The synthetic samples at the final iteration constitute the DP synthetic dataset.

**Theoretical analysis.** Since we only modify RANDOM_API and VARIATION_API, the *privacy guarantee* and *convergence analysis* are exactly the same as PE (Lin et al., 2023) (more details in App. B).

**Considered simulators.** Existing popular image simulators often provide different levels of access. **Some simulators are open-sourced.** Examples include KUBRIC (Greff et al., 2022), a Blender-based renderer for multi-object images/videos; 3D TEAPOT (Lin et al., 2020b; Eastwood & Williams, 2018), an OpenDR-based renderer for teapot images; and PYTHON-AVATAR (Escartín, 2021), a rule-based generator for avatars. However, the assets (e.g., 3D models) used in these renderers are often proprietary. Therefore, **many simulator works choose to release only the generated datasets without the simulator code.** Examples include the FACE SYNTHETICS (Wood et al., 2021) and the DIGIFACE-1M (Bae et al., 2023) datasets, both generated using Blender-based renderers for human faces. In § 3.2 and 3.3, we discuss the design for simulators with code access and data access, respectively.

## 3.2 SIM-PE WITH SIMULATOR ACCESS

While different simulators have very different programming interfaces, most of them can be abstracted in the same way. Given a set of $q$ *numerical* parameters $\phi_1, \ldots, \phi_q$ and $p$ *categorical* parameters $\xi_1, \ldots, \xi_p$ where $\phi_i \in \Phi_i$ and $\xi_i \in \Xi_i$, the simulator $\mathcal{S}$ generates an image $\mathcal{S}(\xi_1, \ldots, \xi_p, \phi_1, \ldots, \phi_q)$. Here, *numerical parameters* refer to those with meaningful ordering (e.g., 1 is closer to 2 than to 3), whether discrete (e.g., $\in \{0, 1, 2\}$) or continuous (e.g., $\in [0, 2]$). In contrast, *categorical parameters* are those *without* inherent ordering, or where even a small change in the parameter can cause a large, unrelated change in the output. For example, for face image renders (Wood et al., 2021; Bae et al., 2023), $\phi_i$s could be the angle of the face and the strength of lighting, and $\xi_i$s could be the ID of the 3D human face model and the ID of the hairstyle.[1]

For RANDOM_API, we simply draw each parameter randomly from its corresponding feasible set:

$$\text{RANDOM\_API} = \mathcal{S}(\xi_1, \ldots, \xi_p, \phi_1, \ldots, \phi_q), \tag{1}$$
$$\text{where } \xi_i \sim \text{Uniform}(\Xi_i) \text{ and } \phi_i \sim \text{Uniform}(\Phi_i).$$

Here, Uniform $(S)$ denotes drawing a sample uniformly at random from the set $S$.

For VARIATION_API, we generate variations by perturbing the input image parameters. For numerical parameters $\phi_i$, we simply add noise. However, for categorical parameters $\xi_i$, where no natural ordering exists among feasible values in $\Xi_i$, adding noise is not applicable. Instead, we re-draw the parameter from the entire feasible set $\Xi_i$ with a certain probability. Formally, it is defined as

$$\text{VARIATION\_API}(\mathcal{S}(\xi_1, \ldots, \xi_p, \phi_1, \ldots, \phi_q)) = \mathcal{S}(\xi_1', \ldots, \xi_p', \phi_1', \ldots, \phi_q'), \tag{2}$$

where $\phi_i' \sim \text{Uniform}([\phi_i - \alpha, \phi_i + \alpha] \cap \Phi_i)$ and $\xi_i' = \begin{cases} \text{Uniform}(\Xi_i), & \text{with probability } \beta \\ \xi_i, & \text{with probability } 1 - \beta \end{cases}$.

Here, $\alpha$ and $\beta$ control the degree of variation. At one extreme, when $\alpha = \infty$ and $\beta = 1$, VARIATION_API completely discards the information of the input sample and reduces to RANDOM_API. Conversely, when $\alpha = \beta = 0$, VARIATION_API outputs the input sample unchanged.

## 3.3 SIM-PE WITH SIMULATOR-GENERATED DATA

We assume a simulator-generated dataset of $m$ samples, $S_{\text{sim}} = \{z_1, \ldots, z_m\}$. The goal is to select $N_{\text{syn}}$ of them to form the DP synthetic dataset $S_{\text{syn}}$. Before introducing our solution, we discuss why two straightforward approaches fall short.

**Baseline 1: Applying DP_NN_HISTOGRAM on $S_{\text{syn}}$.** One immediate solution is to apply DP_NN_HISTOGRAM in PE (Alg. 2) by treating $S_{\text{sim}}$ as the generated set $S$. In other words, each private sample votes for its nearest neighbor in $S_{\text{sim}}$, and the final histogram, aggregating all votes,

---

[1]For well-documented simulators, obtaining the list of parameters is straightforward. For example, PYTHON-AVATAR, used in § 4.2, lists its parameters in the README. Alternatively, one can use the approach in § 3.3, which does not require explicit parameter identification.

is privatized with Gaussian noise. We then draw samples from $S_{\text{sim}}$ according to the privatized histogram (i.e., Line 8 in Alg. 1) to obtain $S_{\text{syn}}$.

However, the size of the simulator-generated dataset (i.e., $m$) is typically very large (e.g., 1.2 million in Bae et al. (2023)), and the total amount of added Gaussian noise grows with $m$. This means that the resulting histogram suffers from a low signal-to-noise ratio, leading to poor fidelity in $S_{\text{syn}}$.

**Baseline 2: Applying DP_NN_HISTOGRAM on cluster centers of $S_{\text{syn}}$.** To improve the signal-to-noise ratio of the histogram, one solution is to have private samples vote on the cluster centers of $S_{\text{sim}}$ instead of the raw samples. Specifically, we first cluster the samples in $S_{\text{sim}}$ into $N_{\text{cluster}}$ clusters with centers $\{w_1, \ldots, w_{N_{\text{cluster}}}\}$ and have private samples vote on these centers rather than individual samples in $S_{\text{sim}}$.[2] Since the number of bins in the histogram decreases from $m$ to $N_{\text{cluster}}$, the signal-to-noise ratio improves. Following the approach of the previous baseline, we then draw $N_{\text{syn}}$ cluster centers (with replacement) based on the histogram and randomly select a sample from each chosen cluster to construct the final $S_{\text{syn}}$.

However, when the total number of samples $m$ is large, each cluster may contain a diverse set of samples, including those both close to and far from the private dataset. While DP voting on clusters improves the accuracy of the DP histogram and helps select better clusters, there remains a risk of drawing unsuitable samples from the chosen clusters.

**Our approach.** Our key insight is that the unavoidable trade-off between DP histogram accuracy and selection precision (clusters vs. individual samples) stems from forcing private samples to consider all of $S_{\text{sim}}$—either directly (baseline 1) or via cluster centers (baseline 2). But this is not necessary: if a sample is far from the private data, its nearest neighbors in $S_{\text{sim}}$ are likely far too (see App. C for experiments). Thus, we can avoid wasting privacy budget on evaluating such samples.

The iterative selection and refinement process in PE naturally aligns with this idea. For each sample $z_i$, we define its nearest neighbors in $S_{\text{sim}}$ as $q_1^i, \ldots, q_m^i$, sorted by closeness, where $q_1^i = z_i$ is the closest. We define RANDOM_API as drawing a random sample from $S_{\text{sim}}$:

$$\text{RANDOM\_API} \sim \text{Uniform}\left(S_{\text{sim}}\right).$$

Since we draw only $N_{\text{syn}}$ samples (instead of all $m$) from RANDOM_API, the DP histogram has a higher signal-to-noise ratio. In the following steps (Lines 6 to 8 in Alg. 1), we discard samples far from the private data indicated by the DP histogram, and apply VARIATION_API only to the remaining as follows:

$$\text{VARIATION\_API}\left(z_i\right) = \text{Uniform}\left(\left\{q_1^i, \ldots, q_\gamma^i\right\}\right),$$

thus avoiding consideration of nearest neighbors of the removed samples (unless they are also nearest neighbors of retained samples). Similar to $\alpha$ and $\beta$ in § 3.2, $\gamma$ controls the degree of variation. At one extreme, when $\gamma = m$, VARIATION_API disregards the input sample and reduces to RANDOM_API. At the other extreme, when $\gamma = 1$, VARIATION_API returns the input sample unchanged.

We verify the effectiveness of our approach over the two baselines in App. F.1.

**Broader applications.** While our main experiments (§ 4) focus on simulator-generated data, the proposed algorithm can be applied to any public dataset. We demonstrate this broader use in App. D.

### 3.4 SIM-PE WITH BOTH SIMULATORS AND FOUNDATION MODELS

As discussed in § 2.1, simulators and foundation models complement each other across different data domains. Moreover, even within a single domain, they excel in different aspects. For example, computer graphics-based face image generators (Bae et al., 2023; Wood et al., 2021) allow controlled diversity in race, lighting, and makeup while mitigating potential biases in foundation models. However, the generated faces may appear less realistic than those produced by SoTA foundation models. Thus, combining the strengths of both methods for DP data synthesis is highly appealing.

Fortunately, PE naturally supports this integration, as RANDOM_API and VARIATION_API work the same for both foundation models and simulators. While there are many ways to combine them, we explore a simple strategy: using simulators in the early PE iterations to generate diverse seed

---

[2]Note that voting in Lin et al. (2023) is conducted in the image embedding space. Here, $w_i$s represent cluster centers in the embedding space, and each private sample uses its image embedding to find the nearest cluster center.

samples, then switching to foundation models in later iterations to refine details and enhance realism. As shown in § 4, this approach outperforms using either simulators or foundation models alone. See App. I.4 for alternative approaches.

## 4 EXPERIMENTS

### 4.1 EXPERIMENTAL SETUP

#### 4.1.1 DATASETS AND SIMULATORS

**Datasets.** Following prior work (Gong et al., 2025), we use three private datasets: **(1) MNIST** (LeCun, 1998), where the image class labels are digits '0'-'9', **(2) CelebA** (Liu et al., 2015), where the image class labels are male and female, and **(3) CIFAR10** (Krizhevsky et al., 2009), which contains 10 classes of natural images. We aim at *conditional generation* for these datasets (i.e., each generated image is associated with the class label). Due to space constraints, CIFAR10 experiments are deferred to App. D.

**Simulators.** To showcase the broad applicability of Sim-PE, we use four diverse simulators.

**(1) Text rendering program.** Generating images with readable text using foundation models is a known challenge (Betker et al., 2023). Simulators can address this gap, as generating images with text through computer programs is straightforward. To illustrate this, we implement our own text rendering program, treating MNIST as the private dataset. Specifically, we use the Python PIL library to render digits as images. There are two categorical and three numerical parameters in total (details in App. G.1). We set the feasible sets of these parameters to be large enough so that the random samples differ significantly from MNIST (see Fig. 2b).

**(2) Computer graphics-based renderer for face images.** Computer graphics-based rendering is widely used in real-world applications such as game development, cartoons, and movie production. This experiment aims to assess whether these advanced techniques can be adapted for DP synthetic image generation via Sim-PE. We use CelebA as the private dataset and a Blender-based face image renderer from Bae et al. (2023) as the API. Since the source code for their renderer is not publicly available, we apply our data-based algorithm from § 3.3 on their released dataset of 1.2 million face images. Note that this renderer may not necessarily represent the SoTA. As visualized in Fig. 3b, the generated faces exhibit various unnatural artifacts and appear less realistic than images produced by SoTA generative models (e.g., Rombach et al. (2022)). Therefore, this experiment serves as a preliminary study, and *the results could potentially improve with more advanced rendering techniques.*

**(3) Rule-based avatar generator.** We further investigate whether Sim-PE remains effective *when the simulator's data significantly differs from the private dataset.* We use CelebA as the private dataset and a rule-based avatar generator (Escartín, 2021) as the API. This simulator has 16 categorical parameters that control attributes of the avatar, including eyes, noses, background colors, skin colors, etc (details in App. G.2). As in Fig. 4b, the generated avatars have a cartoon-like appearance and lack fine-grained details, which contrasts sharply with real human face photos in CelebA.

**(4) Public images.** To demonstrate that Sim-PE (§ 3.3) generalizes beyond simulator-generated data, we also evaluate it using ImageNet as $S_{\text{sim}}$. Owing to space, results are in App. D.

**Class label information from simulators.** In our main experiments, we assume the simulator does not provide class label information (denoted as "**ClassUnavail**"). In App. F.8, we further investigate a setting where the simulator *does* provide class labels along with the generated images, and find that this leads to improved performance. See App. F.8 for details and results.

#### 4.1.2 METRICS AND EVALUATION PIPELINES

We follow the evaluation settings of DPImageBench (Gong et al., 2025), a recent benchmark for DP image synthesis. Specifically, we use two metrics: **(1) FID** (Heusel et al., 2017) as a quality metric and **(2) the accuracy of downstream classifiers** as a utility metric. Specifically, we use the conditional version of PE (App. A), so that each generated images are associated with the class labels (i.e., '0'-'9' digits in MNIST, male vs. female in CelebA). These class labels are the targets for training the classifiers. We employ a strict train-validation-test split and account for the privacy cost of classifier hyperparameter selection. Specifically, we divide the private dataset into disjoint training and validation sets. We then run Sim-PE on the training set to generate synthetic data. Next, we train three classifiers—ResNet (He et al., 2016), WideResNet (Zagoruyko & Komodakis, 2016),

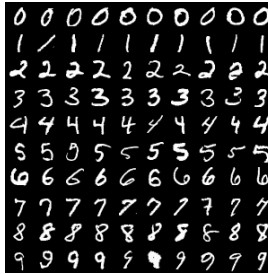 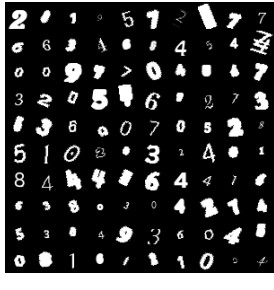 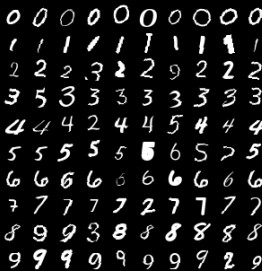

(a) Real (private) images      (b) Simulator-generated images      (c) Sim-PE images ($\epsilon = 10$)

Figure 2: The real and generated images on MNIST under the "**ClassUnavail**" setting. Each row corresponds to one class. The simulator generates images that are very different from the real ones and are from the incorrect classes. Starting from these bad images, Sim-PE can effectively guide the generation of the simulator towards high-quality images with correct classes.

Table 1: Accuracy and FID. The best between PE methods is in **bold**, and the best between all methods is underlined. "Simulator" refers to samples from the simulator's RANDOM_API. Results other than Sim-PE and Simulator are taken from Gong et al. (2025).

(a) Accuracy (%) of downstream classifiers.

| Algorithm | MNIST | | CelebA | |
|---|---|---|---|---|
| | $\epsilon = 1$ | $\epsilon = 10$ | $\epsilon = 1$ | $\epsilon = 10$ |
| DP-MERF | 80.3 | 81.3 | 81.0 | 81.2 |
| DP-NTK | 50.0 | 91.3 | 61.2 | 64.2 |
| DP-Kernel | 94.0 | 93.6 | 83.0 | 83.7 |
| GS-WGAN | 72.4 | 75.3 | 61.4 | 61.5 |
| DP-GAN | 92.4 | 92.7 | 77.9 | 89.2 |
| DPDM | 89.2 | 97.7 | 74.5 | 91.8 |
| PDP-Diffusion | 94.5 | 97.4 | 89.4 | 94.0 |
| DP-LDM (SD) | 78.8 | 94.4 | 84.4 | 89.1 |
| DP-LDM | 44.2 | 95.5 | 85.8 | 92.4 |
| DP-LoRA | 82.2 | 97.1 | 87.0 | 92.0 |
| PrivImage | 94.0 | 97.8 | 90.8 | 92.0 |
| Simulator | 11.6 ($\epsilon = 0$) | | 61.4 ($\epsilon = 0$) | |
| PE | 27.9 | 32.7 | 70.5 | 74.2 |
| Sim-PE (ours) | **89.1** | **93.6** | **80.0** | **82.5** |

(b) FID of synthetic images.

| Algorithm | MNIST | | CelebA | |
|---|---|---|---|---|
| | $\epsilon = 1$ | $\epsilon = 10$ | $\epsilon = 1$ | $\epsilon = 10$ |
| DP-MERF | 113.7 | 106.3 | 176.3 | 147.9 |
| DP-NTK | 382.1 | 69.2 | 350.4 | 227.8 |
| DP-Kernel | 33.7 | 38.9 | 140.3 | 128.8 |
| GS-WGAN | 57.0 | 47.7 | 611.8 | 290.0 |
| DP-GAN | 82.3 | 30.3 | 112.5 | 31.7 |
| DPDM | 36.1 | 4.4 | 153.99 | 28.8 |
| PDP-Diffusion | 8.9 | 3.8 | 17.1 | 8.1 |
| DP-LDM (SD) | 31.9 | 18.7 | 46.2 | 24.1 |
| DP-LDM | 155.2 | 99.1 | 124.1 | 40.4 |
| DP-LoRA | 112.8 | 95.4 | 53.3 | 32.2 |
| PrivImage | 7.6 | 2.3 | 11.4 | 11.3 |
| Simulator | 86.2 ($\epsilon = 0$) | | 37.2 ($\epsilon = 0$) | |
| PE | 48.8 | 45.3 | **23.4** | 22.0 |
| Sim-PE (ours) | **20.7** | **9.4** | 24.7 | **20.8** |

and ResNeXt (Xie et al., 2017)—on the synthetic data and evaluate their accuracy on the validation set. Since the validation set is part of the private data, we use the Report Noisy Max algorithm (Dwork et al., 2014) to select the best classifier checkpoint across all epochs of all three classifiers. Finally, we report the accuracy of this classifier on the test set. This procedure ensures that the reported accuracy is not inflated due to train-test overlap or DP violations in classifier hyperparameter tuning. See App. I.2 for further discussion on the rationale behind our choice of metrics.

Following Gong et al. (2025), we set DP parameter $\delta = 1/(N_{\text{priv}} \cdot \log N_{\text{priv}})$, where $N_{\text{priv}}$ is the number of samples in the private dataset, and $\epsilon = 1$ or 10.

### 4.1.3 BASELINES

We compare Sim-PE with 12 SoTA DP image synthesizers reported in Gong et al. (2025), including DP-MERF (Harder et al., 2021a), DP-NTK (Yang et al., 2023), DP-Kernel (Jiang et al., 2023), GS-WGAN (Chen et al., 2020), DP-GAN (Xie et al., 2018), DPDM (Dockhorn et al., 2023), PDP-Diffusion (Ghalebikesabi et al., 2023b), DP-LDM (Liu et al., 2024b), DP-LoRA (Tsai et al., 2024), PrivImage (Li et al., 2024), and PE with foundation models (Lin et al., 2023). Except for PE, all other baselines require model training. When experimenting with simulator-generated data, we additionally compare Sim-PE against the two baselines introduced in § 3.3.

**Note:** Different methods rely on different prior knowledge. For instance, many baselines use pre-trained models or public data from similar distributions (see Tab. 2 of Gong et al. (2025)), whereas Sim-PE does not. Instead, Sim-PE leverages simulators, which are not used by others. **As such, Sim-PE represents a new evaluation setting, and baseline results serve primarily to contextualize this new paradigm and inspire future work, rather than an apple-to-apple comparison.**

### 4.2 SIM-PE WITH SIMULATOR ACCESS

In this section, we evaluate Sim-PE with a text rendering program on MNIST dataset. The results are shown in Tab. 1 and Fig. 2. The key takeaway messages are:

(a) Real (private) images     (b) Simulator-generated images     (c) Sim-PE images ($\epsilon = 10$)

Figure 3: The real and generated images on CelebA. The top rows correspond to the "female" class, and the bottom rows correspond to the "male" class. The simulator generates images with incorrect classes. However, starting from these misclassified images, Sim-PE effectively selects those that better match the correct class.

Table 2: Accuracy (%) of classifiers trained on synthetic images and FID of synthetic images on CelebA. The best results are highlighted in bold. Using a combination of both (weak) simulators and foundation models outperforms using either one alone.

| Algorithm | FID ↓ | | Classification Acc. ↑ | |
|---|---|---|---|---|
| | $\epsilon = 1$ | $\epsilon = 10$ | $\epsilon = 1$ | $\epsilon = 10$ |
| PE with foundation models | 23.4 | 22.0 | 70.5 | 74.2 |
| PE with weak simulators (i.e., Sim-PE) | 101.4 | 99.5 | 62.6 | 63.2 |
| PE with both | **15.0** | **11.9** | **72.7** | **78.1** |

**Sim-PE effectively guides the simulator to generate high-quality samples.** As shown in Fig. 2b, without any information from the private data or guidance from Sim-PE, the simulator initially produces poor-quality images with incorrect digit sizes, rotations, and stroke widths. These low-quality samples serve as the starting point for Sim-PE (via RANDOM_API). Through iterative refinement and private data voting, Sim-PE gradually optimizes the simulator parameters, ultimately generating high-quality MNIST samples, as illustrated in Fig. 2c.

Quantitative results in Tab. 1 further support this. Without private data guidance, the simulator often generates digits from incorrect classes, yielding a classifier accuracy of just 11.6%–near random guess. In contrast, Sim-PE boosts accuracy to around 90%. FID scores also confirm that Sim-PE produces images more similar to real data.

**Sim-PE can improve the performance of PE by a large margin.** The PE baseline (Lin et al., 2023) uses a diffusion model pre-trained on ImageNet, which primarily contains natural object images (e.g., plants, animals, cars). Since MNIST differs significantly from such data, PE, as a training-free method, struggles to generate meaningful MNIST-like images. Most PE-generated images lack recognizable digits (see Gong et al. (2025)), resulting in a classification accuracy of only $\sim 30\%$ (Tab. 1a). By leveraging a simulator better suited for this domain, Sim-PE achieves much better results, tripling the classification accuracy and reducing the FID by 80% at $\epsilon = 10$.

**Sim-PE achieves competitive results among SoTA methods.** When the foundation model or public data differs significantly from the private data, training-based baselines can still adapt the model to the private data distribution by updating its weights, whereas PE cannot. This limitation accounts for the substantial performance gap between PE and other methods. Specifically, PE records the lowest classification accuracy among all 12 methods (Tab. 1a). By leveraging domain-specific simulators, Sim-PE substantially narrows this gap, achieving classification accuracy within 5.4% and 4.2% of the best-performing method for $\epsilon = 1$ and $\epsilon = 10$, respectively.

## 4.3 SIM-PE WITH SIMULATOR-GENERATED DATA

We evaluate Sim-PE using a generated dataset from a computer graphics-based renderer on the CelebA dataset. The results, presented in Tab. 1 and Fig. 3, highlight the following key takeaways:

**Sim-PE selects samples that better match target classes.** Without private data, the simulator naturally generates images with incorrect labels (Fig. 3b). As a result, a gender classifier trained on this data achieves at most 61.4% accuracy–the majority class (female) rate. Sim-PE iteratively refines selection from this noisy pool, ultimately choosing samples more aligned with target classes (Fig. 3c), boosting accuracy by up to 21.1% (Tab. 1a).

**Sim-PE maintains the strong data quality of PE.** As shown in Tab. 1b, Sim-PE and PE achieve similar FID. Unlike in MNIST (§ 4.2) where Sim-PE brought large gains, the modest improvement on CelebA stems from two factors. First, PE with foundation models already ranks 3rd in FID, leaving little room to improve. Second, Sim-PE here only selects from a fixed simulator-generated dataset. As seen in Fig. 3, these images differ from real CelebA (e.g., having larger faces), and it is impossible for Sim-PE to correct such discrepancies. Having access to simulator code, as in § 4.2, could help alleviate such errors by allowing parameter adjustments. A hybrid approach combining foundation models and simulators, as we will explore next, may also offer further gains.

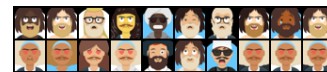

(a) Real (private) images   (b) Simulator-generated images   (c) Sim-PE images ($\epsilon = 10$)

Figure 4: The real and generated images on CelebA. The simulator is a weak rule-based avatar generator (Escartín, 2021) significantly different from the real dataset. The top rows correspond to the "female" class, and the bottom rows correspond to the "male" class. The simulator generates images with incorrect classes. Sim-PE tends to generate faces with long hair for the female class and short hair for the male class (correctly), but the generated images have mode collapse issues.

## 4.4 SIM-PE WITH BOTH SIMULATORS AND FOUNDATION MODELS

In this section, we examine Sim-PE under weak simulators. As in the previous section, we use CelebA as the private dataset, but replace the simulator with a rule-based cartoon avatar generator (Escartín, 2021). As shown in Fig. 4b, the generated avatars differ significantly from real images.

**Sim-PE with weak simulators still learns useful features.** From Tab. 2, we observe that downstream classifiers trained on Sim-PE with weak simulators achieve poor classification accuracy. However, two interesting results emerge: **(1)** Despite the significant difference between avatars and real face images, Sim-PE still captures certain characteristics of the two classes correctly. Specifically, Sim-PE tends to generate faces with long hair for the female class and short hair for the male class (Fig. 4c). **(2)** Although the FID of Sim-PE is quite poor (Tab. 2), they still outperform many baselines (Tab. 1b). This can be explained by the fact that, as shown in Gong et al. (2025), high DP noise makes the training of many baselines unstable. This results in images with noisy patterns, non-face images, or significant mode collapse, particularly for DP-NTK, DP-Kernel, and GS-WGAN. In contrast, Sim-PE is training-free, and thus it avoids these issues. See App. E for more analysis.

Next, we explore the feasibility of using PE with both foundation models and the weak avatar simulator (§ 3.4). The results are shown in Tab. 2.

**Sim-PE benefits from utilizing simulators and foundation models together.** We observe that using both simulators and foundation models yields the best results in terms of both FID and classification accuracy. This result is intuitive: the foundation model, pre-trained on the diverse ImageNet dataset, has a low probability of generating a face image through RANDOM_API. While avatars are quite different from CelebA, they retain the correct image layout, such as facial boundaries, eyes, nose, etc. Using these avatars as seed samples for variation allows the foundation model to focus on images closer to real faces, rather than random, unrelated patterns.

Unlike other SoTA methods that are tied to a specific data synthesizer, this result suggests that PE is a promising framework that can easily combine the strengths of multiple types of data synthesizers.

## 4.5 FURTHER ANALYSES

**Number of iterations.** Like PE, Sim-PE achieves good results in a few (e.g., 6) iterations (Fig. 8).

**Computation cost.** Since simulators could be much cheaper to run than foundation models, Sim-PE could be much more efficient than PE. For instance, on MNIST, each PE's API call takes over *2400 GPU seconds*, whereas Sim-PE takes less than *30 CPU seconds*–an 80x speedup, not to mention the lower cost of CPU than GPU. See App. H for detailed results.

**Hyperparameter sensitivity.** We conduct ablation studies in App. F. Overall, Sim-PE is robust to simulator parameter ranges, Sim-PE hyperparameters ($\alpha, \beta, \gamma$), and the timing of switching from simulators to foundation models. We also study the effect of sample size and distribution alignment.

## 5 LIMITATIONS AND FUTURE WORK

In this paper, we demonstrate the potential of the PE framework for utilizing powerful simulators in DP image synthesis. While the results are promising, several important questions remain open:
**(1)** Since Sim-PE (in § 3.3) can be applied to any public dataset (App. D), it could support applications such as pre-training data selection for private fine-tuning (Yu et al., 2023; Li et al., 2024). **(2)** This paper applies Sim-PE on images. In domains like networking and systems, simulators are more common than foundation models, and Sim-PE could offer even greater potential. **(3)** Sim-PE for image synthesis is still outperformed by the best baseline. Exploring better ways to combine simulators and foundation models could further push the limits of the PE framework. **(4)** It remains an open question how to effectively apply PE in domains where suitable foundation models, simulators, and public datasets are all unavailable.

## REPRODUCIBILITY STATEMENT

The hyperparameters are reported in App. G, and the code for reproducing the experiments is provided in the supplementary materials. The code will be open-sourced.

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

APPENDIX CONTENTS

# A PRIVATE EVOLUTION

Alg. 1 presents the Private Evolution (PE) algorithm, reproduced from Lin et al. (2023). This algorithm represents the conditional version of PE, where each generated image is associated with a class label. It can be interpreted as running the unconditional version of PE separately for each class (Line 2).

---

**Algorithm 1:** Private Evolution (PE)

---

**Input:** The set of private classes: $C$   ($C = \{0\}$ if for unconditional generation)

Private samples: $S_{\text{priv}} = \{(x_i, y_i)\}_{i=1}^{N_{\text{priv}}}$, where $x_i$ is a sample and $y_i \in C$ is its label
Number of iterations: $T$
Number of generated samples: $N_{\text{syn}}$   (assuming $N_{\text{syn}} \bmod |C| = 0$)
Noise multiplier for DP Nearest Neighbors Histogram: $\sigma$
Threshold for DP Nearest Neighbors Histogram: $H$

1   $S_{\text{syn}} \leftarrow \emptyset$
2   **for** $c \in C$ **do**
3     $private\_samples \leftarrow \{x_i | (x_i, y_i) \in S_{\text{priv}} \text{ and } y_i = c\}$
4     $S_0 \leftarrow \mathsf{RANDOM\_API}\,(N_{\text{syn}}/|C|)$
5     **for** $t \leftarrow 1, \ldots, T$ **do**
6       $histogram_t \leftarrow \mathsf{DP\_NN\_HISTOGRAM}\,(private\_samples, S_{t-1}, \sigma, H)$      `// See` Alg. 2
7       $\mathcal{P}_t \leftarrow histogram_t/\text{sum}(histogram_t)$    `//` $\mathcal{P}_t$ `is a distribution on` $S_t$
8       $S'_t \leftarrow$ draw $N_{\text{syn}}/|C|$ samples with replacement from $\mathcal{P}_t$    `//` $S'_t$ `is a multiset`
9       $S_t \leftarrow \mathsf{VARIATION\_API}\,(S'_t)$
10    $S_{\text{syn}} \leftarrow S_{\text{syn}} \cup \{(x, c) | x \in S_T\}$
11 **return** $S_{\text{syn}}$

---

---

**Algorithm 2:** DP Nearest Neighbors Histogram (DP_NN_HISTOGRAM)

---

**Input**  : Private samples: $S_{\text{priv}}$

Generated samples: $S = \{z_i\}_{i=1}^n$
Noise multiplier: $\sigma$
Threshold: $H$
Distance function: $d\,(\cdot, \cdot)$

**Output:** DP nearest neighbors histogram on $S$

1   $histogram \leftarrow [0, \ldots, 0]$
2   **for** $x_{\text{priv}} \in S_{\text{priv}}$ **do**
3     $i = \arg\min_{j \in [n]} d\,(x_{\text{priv}}, z_j)$
4     $histogram[i] \leftarrow histogram[i] + 1$
5   $histogram \leftarrow histogram + \mathcal{N}\,(0, \sigma I_n)$       `// Add noise to ensure DP`
6   $histogram \leftarrow \max\,(histogram - H, 0)$    `// 'max', '-' are element-wise`
7   **return** $histogram$

---

**Embedding.** Computing the sample distance in Line 3 is performed in an embedding space, following (Lin et al., 2023). The VARIATION_API in § 3.2 also operates in an embedding space. For our image experiments, we use embeddings produced by a pre-trained model (Inception), and we assume that the data used to train this embedding model (ImageNet) is public and independent of the private dataset.

# B THEORETICAL ANALYSIS

**Convergence analysis.** Since Sim-PE only changes PE's RANDOM_API and VARIATION_API, the original convergence analysis (App. E in Lin et al. (2023)) remains valid for Sim-PE. Specifically:

- The analysis assumes that RANDOM_API generates samples within a ball of diameter $D$ covering the private samples. Changing RANDOM_API affects $D$ but not the overall analysis procedure.
- The analysis assumes that VARIATION_API draws samples from a Gaussian distribution—an essential assumption for any tractable analysis involving complex, unknown foundation models and simulators. Under this assumption, modifying VARIATION_API does not impact the analysis procedure.

**Privacy analysis.** Since Sim-PE does not change the way PE utilizes private data, the privacy analysis of PE (Lin et al., 2023) applies to Sim-PE. We repeat the privacy analysis in PE below for completeness.

The only step that directly accesses private data is DP_NN_HISTOGRAM (Line 6 in Alg. 1, detailed in Alg. 2). Let us analyze its privacy cost first:

- Each private sample contributes only one vote (Line 5 in Alg. 2).
- Adding or removing a single sample alters the resulting histogram by at most 1 in the $l_2$ norm—so the sensitivity is 1.
- Line 5 adds i.i.d. Gaussian noise with standard deviation $\sigma$ to each histogram bin. This corresponds to the standard Gaussian mechanism (Dwork et al., 2014) with noise multiplier $\sigma$ (i.e., $\sigma/1$).

Now, consider the entire Sim-PE procedure (Alg. 1).

*(1) Let's first consider the procedure for one class $c$ (Line 3-Line 10).* All steps other than DP_NN_HISTOGRAM do not access private data and simply post-process results from earlier DP steps. Therefore, Alg. 1 can be interpreted as $T$ *adaptive compositions* of Gaussian mechanisms.

According to Corollary 3.3 in Dong et al. (2022),[3] $T$ *compositions of Gaussian mechanisms* with noise multiplier $\sigma$ can be viewed as a *single Gaussian mechanism* with effective noise multiplier $\sigma/\sqrt{T}$.

Thus, the final privacy cost reduces to computing the $\varepsilon$ and $\delta$ of a standard Gaussian mechanism with noise $\sigma/\sqrt{T}$. We use the tight results from Balle & Wang (2018) for this computation: the mechanism is $(\varepsilon, \delta)$-differentially private if

$$\Phi\left(\frac{\sqrt{T}}{2\sigma} - \frac{\varepsilon\sigma}{\sqrt{T}}\right) - e^\varepsilon\,\Phi\left(-\frac{\sqrt{T}}{2\sigma} - \frac{\varepsilon\sigma}{\sqrt{T}}\right) \le \delta$$

where $\Phi$ denotes the Gaussian cumulative distribution function (CDF). Given $\delta$ and $\sigma$, we can use the above equation to solve for $\varepsilon$.

*(2) Now, let's first consider the full algorithm involving multiple classes.* Since the private samples from different classes are disjoint, due to *parallel composition property* (McSherry, 2009), the privacy cost remains the same as above. See Lin et al. (2023) for an alternative analysis that gives the same conclusion.

## C  JUSTIFYING THE METHODOLOGICAL DESIGN OF SIM-PE WITH SIMULATOR-GENERATED DATA

In this section, we provide experimental evidence to support the claim in § 3.3: "If we already know that a sample $z_i$ is far from the private dataset, then its nearest neighbors in $S_{\text{sim}}$ are also likely to be far from the private dataset."

Consider Sim-PE's synthetic dataset $S_t$ from the $t$-th iteration. For each private sample $x_i$ in the private dataset $S_{\text{priv}}$, we find its nearest sample in $S_t$ as $z_i$. $S(selected)_t = \{z_i\}_{i=1}^{N_{\text{priv}}}$ contains samples close to $S_{\text{priv}}$ (chosen by DP_NN_HISTOGRAM (Alg. 2) in PE under non-DP settings). Conversely, the remaining samples, $S(unselected)_t = S_t \setminus S(selected)_t$ are farther from $S_{\text{priv}}$.

Now, we need to confirm that $S(unselected)_t$'s nearest neighbors in $S_{\text{sim}}$ are farther away from the private dataset $S_{\text{priv}}$ than $S(selected)_t$. We define:

---

[3]Please see their arXiv version https://arxiv.org/pdf/1905.02383.

- $z(Y)_i$ as the nearest neighbor of the private sample $x_i$ in $S(Y)_t$, for $Y \in \{selected, unselected\}$
- $q(Y)_j^i$ as the $j-th$ nearest neighbor of $z(Y)_i$ in $S_{\text{sim}}$
- $FID(Y)$ as the FID between $S_{priv}$ and $\left\{q(Y)_j^i\right\}_{i=1,j=1}^{N_{\text{priv}},\gamma}$.

Fig. 5 shows that $FID(selected)$ is smaller (better) than $FID(unselected)$ across most PE iterations, supporting our claim.

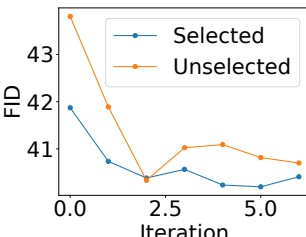

Figure 5: FID of the nearest neighbors of the selected and unselected samples.

## D SIM-PE WITH CIFAR10 AS THE PRIVATE DATASET AND IMAGENET AS THE SIMULATOR-GENERATED DATASET

In this section, we apply the algorithm in § 3.3 on the setting where CIFAR10 is treated as the private dataset and ImageNet is treated as the simulator-generated dataset. We conduct this experiment for the following reasons:

- In § 3.3, we discuss that Sim-PE can apply to any public dataset beyond simulator-generated data. This experiment provides evidence for that claim.
- The original PE approach on CIFAR10 (Lin et al., 2023) uses a foundation model pretrained on ImageNet, which aligns with the simulator data in this experiment. Comparing their performance removes the impact of distribution alignment and highlights the differences in the RANDOM_API and VARIATION_API designs of PE and Sim-PE.

**Experiment settings.** We use $\epsilon = 10$. The downstream task is classifying images into the 10 classes in CIFAR10, following DPImageBench (Gong et al., 2025) and the original PE paper (Lin et al., 2023).

**Results.** Sim-PE achieves FID=8.76, and acc=70.06%. In comparison, PE with a foundation model achieves similarly with FID=9.2 and acc=75.3%. These results suggest that Sim-PE and PE can be competitive when both suitable simulators and foundation models are available for the private dataset. In addition, the results are comparable to state-of-the-art training-based methods, demonstrating that Sim-PE is a promising framework.

## E MORE ANALYSIS ON SIM-PE WITH WEAK SIMULATORS

In Fig. 4c, we see that Sim-PE's generated images have many duplicates. In this section, we conduct more analysis on this phenomenon.

Fig. 6 shows that FID between generated images and the private dataset decreases (improves) across iterations, confirming that generated images better align with the private distribution over iterations. However, the number of unique samples in the generated dataset drops significantly (Fig. 7), leading to duplicates in Fig. 4c. This may be due to the simulator being too weak, generating most images far from the private dataset. As a result, Sim-PE converges to a small set of good images.

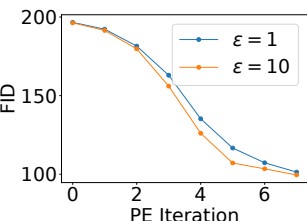

Figure 6: Sim-PE (using DIGIFACE-1M)'s FID on CelebA improves over the course of the PE iterations.

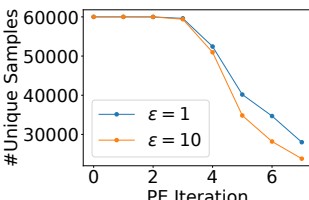

Figure 7: The number of unique generated samples in each iteration of Sim-PE (using DIGIFACE-1M) on CelebA.

## F ADDITIONAL ABLATION STUDIES

### F.1 DATA SELECTION ALGORITHMS

In § 3.3, we discussed two simple alternatives for simulator data selection. The comparison is in Tab. 3. We see that Sim-PE with iterative data selection outperforms the baselines on most metrics, validating the intuition outlined in § 3.3. However, the clustering approach used in the second baseline still has merit, as it results in a better FID for $\epsilon = 10$. This idea is orthogonal to the design of Sim-PE and could potentially be combined for further improvement. We leave this exploration to future work.

Table 3: Accuracy (%) of classifiers trained on synthetic images and FID of synthetic images on CelebA. The best results are highlighted in bold. Sim-PE outperforms the baselines in most metrics.

| Algorithm | FID ↓ | | Classification Acc. ↑ | |
|---|---|---|---|---|
| | $\epsilon = 1$ | $\epsilon = 10$ | $\epsilon = 1$ | $\epsilon = 10$ |
| DP_NN_HISTOGRAM on $S_{\text{syn}}$ | 36.2 | 29.3 | 61.5 | 71.9 |
| DP_NN_HISTOGRAM on cluster centers of $S_{\text{syn}}$ | 26.4 | **18.3** | 74.7 | 77.7 |
| Sim-PE | **24.7** | 20.8 | **80.0** | **82.5** |

### F.2 PERFORMANCE ACROSS ITERATIONS

Fig. 8 shows that both the FID and the downstream classifier's accuracy generally improve as PE progresses. This confirms that PE's iterative data refinement process is effective when combined with simulators.

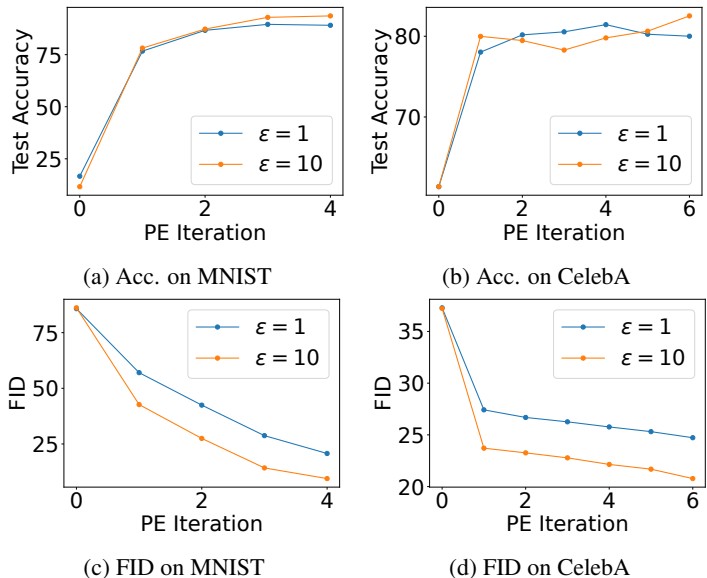

Figure 8: Sim-PE's FID and accuracy generally improve over the course of the iterations.

### F.3 SENSITIVITY ANALYSIS OF $\alpha, \beta, \gamma$

These hyperparameters (defined in § 3.2 and 3.3) control the variation degree in VARIATION_API. Higher values induce larger variations. **These are the only new hyperparameters introduced in Sim-PE.**

Following the PE paper (Lin et al., 2023), we gradually decrease them across iterations. The idea is that larger variations at the beginning help Sim-PE/PE explore and find good seeds, while smaller variations are needed later for convergence. In our experiments, most are set to follow an arithmetic or geometric sequence across iterations (App. G) based on heuristics. In the following, we modify their values: for each hyperparameter, we either fix it to the largest or smallest value in its base sequence. Figs. 9 and 10 show that:

- Setting these hyperparameters to a fixed *large* value is not ideal. For instance, setting a large text variation throughout greatly hurts FID and accuracy. Conversely, using a fixed *small* value has a smaller impact.
- The default parameters do not always yield the best result (e.g., setting $\gamma$ to a fixed small value improves FID). This suggests that our main results in the paper could be improved with better hyperparameters.

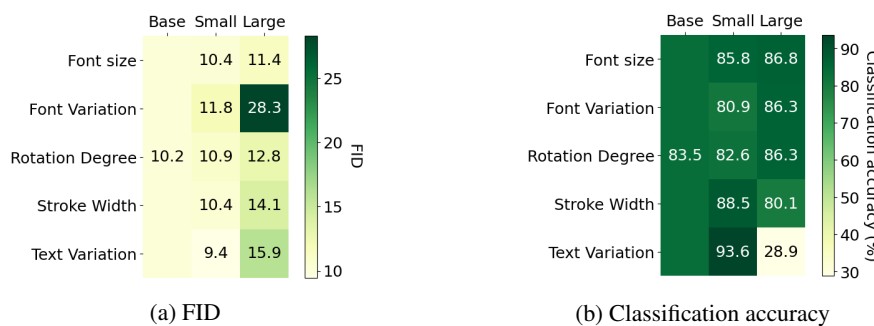

(a) FID                                    (b) Classification accuracy

Figure 9: Accuracy and FID on MNIST with different $\alpha, \beta$ schedules ($\epsilon = 10$). The "base" parameter schedule across iterations is: (a) font size's $\alpha = [5, 4, 3, 2]$, (b) rotation degree's $\alpha = [9, 7, 5, 3]$, (c) stroke width's $\alpha = [1, 1, 0, 0]$, (d) font's $\beta = [0.8, 0.4, 0.2, 0.0]$, (e) text's $\beta = [0.8, 0.4, 0.2, 0.0]$. "Small" ("large") means fixing $\alpha/\beta$ to be the smallest (largest) value across iterations. For example, the unit under "font variation" and "large" indicates the results when we set font's $\beta = [0.8, 0.8, 0.8, 0.8]$.

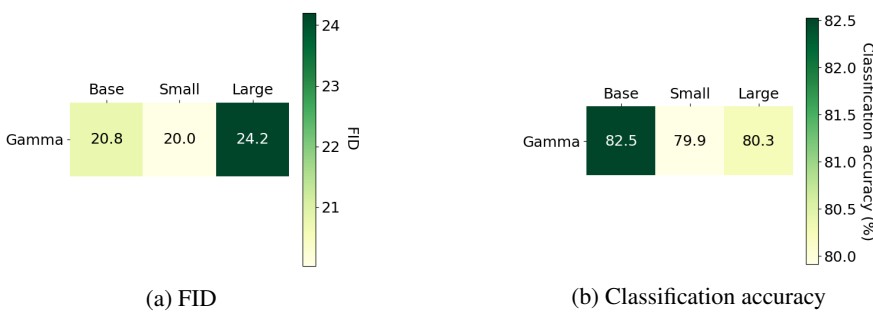

(a) FID                                    (b) Classification accuracy

Figure 10: Accuracy and FID on CelebA using simulator-generated data with different $\gamma$ schedules ($\epsilon = 10$). The "base" parameter schedule across iterations is $[1000, 500, 200, 100, 50, 20]$. "Small" ("large") means fixing $\gamma$ to be the smallest (largest) value across iterations. For example, the unit under "large" indicates the results when we set $\gamma = [1000, 1000, 1000, 1000, 1000, 1000]$.

### F.4 WHEN TO SWITCH FROM THE SIMULATOR TO THE FOUNDATION MODEL

In § 4.4, the foundation model was introduced after iteration 1. We vary it to be either earlier (after iteration 0, using only simulators for the initial samples) or later (after iteration 2). The result is in Tab. 4. We can see that:

- Our reported main results can be improved with better parameters. For instance, starting the foundation model after iteration 0 improves accuracy to 74.3% (+1.6%) and reduces FID to 14.2 (-0.8) at $\varepsilon = 1$.
- Nevertheless, the results across different schedules are relatively close. In most cases, they outperform using either the foundation model or the simulator alone (see Tab. 2), reinforcing the conclusions in § 4.4.

| Diffusion Start | FID ($\varepsilon = 1$) | Acc ($\varepsilon = 1$) | FID ($\varepsilon = 10$) | Acc ($\varepsilon = 10$) |
|---|---|---|---|---|
| After iter 0 | 14.2 | 74.3 | 13.2 | 73.6 |
| After iter 1 | 15.0 | 72.7 | 11.9 | 78.1 |
| After iter 2 | 16.0 | 73.6 | 14.2 | 76.5 |

Table 4: Comparison of FID and accuracy with different diffusion start iterations for $\varepsilon = 1$ and $\varepsilon = 10$.

### F.5 RANGE OF SIMULATOR PARAMETERS

We vary the range of all numerical parameters in the MNIST experiment $\varepsilon = 1$, including font size, rotation, and stroke width. Results across all Sim-PE iterations are presented.

- **Font size.** In our main experiments, we used the range [10, 30). We evaluate two alternatives: [15, 25) (narrower) and [5, 35) (wider). See Tab. 5 for the results.
- **Rotation degree.** In our main experiments, we used the range [–30, 30]. We evaluate two alternatives: [–15, 15] (narrower) and [–60, 60] (wider). See Tab. 6 for the results.
- **Stroke width.** In our main experiments, we used the range [0, 2]. We evaluate two alternatives: [0, 1] (narrower) and [0, 3] (wider). See Tab. 7 for the results.

We can see from the above three experiments that:

- For font size and rotation, varying the parameter range has minimal effect on accuracy or FID across iterations. In contrast, a suboptimal stroke width range ([0, 3]) leads to poor initial samples (FID 109.1), but Sim-PE gradually improves quality, narrowing the final FID gap to the best setting to 4.3.
- Regardless of the specific parameter range chosen, the overall trend is consistent: Sim-PE consistently improves the synthetic data over iterations and outperforms standard PE significantly (see Table 1). These are the core messages of the paper, and they hold even when the parameter ranges are varied.

| Range | Metric | Iter=0 | Iter=1 | Iter=2 | Iter=3 | Iter=4 |
|---|---|---|---|---|---|---|
| [15, 25) | Accuracy | 13.6 | 71.6 | 81.4 | 87.4 | 84.8 |
| | FID | 88.1 | 61.8 | 45.3 | 31.1 | 22.4 |
| [10, 30) | Accuracy | 16.6 | 76.7 | 86.7 | 89.5 | 89.1 |
| | FID | 85.8 | 57.0 | 42.4 | 28.7 | 20.7 |
| [5, 35) | Accuracy | 13.3 | 66.0 | 82.8 | 90.6 | 86.2 |
| | FID | 87.2 | 58.2 | 41.4 | 28.7 | 20.2 |

Table 5: Effect of font size range on accuracy and FID.

| Range | Metric | Iter=0 | Iter=1 | Iter=2 | Iter=3 | Iter=4 |
|---|---|---|---|---|---|---|
| [–15, 15] | Accuracy | 14.5 | 75.7 | 86.0 | 88.3 | 87.7 |
| | FID | 83.1 | 56.7 | 42.3 | 28.6 | 19.7 |
| [–30, 30] | Accuracy | 16.6 | 76.7 | 86.7 | 89.5 | 89.1 |
| | FID | 85.8 | 57.0 | 42.4 | 28.7 | 20.7 |
| [–60, 60] | Accuracy | 13.3 | 72.3 | 83.0 | 88.5 | 86.3 |
| | FID | 86.6 | 56.0 | 40.9 | 27.5 | 19.4 |

Table 6: Effect of rotation degree range on accuracy and FID.

| Range | Metric | Iter=0 | Iter=1 | Iter=2 | Iter=3 | Iter=4 |
|-------|--------|--------|--------|--------|--------|--------|
| [0, 1] | Accuracy | 12.8 | 70.0 | 87.6 | 91.5 | 91.3 |
|        | FID | 69.0 | 48.7 | 35.3 | 25.1 | 18.1 |
| [0, 2] | Accuracy | 16.6 | 76.7 | 86.7 | 89.5 | 89.1 |
|        | FID | 85.8 | 57.0 | 42.4 | 28.7 | 20.7 |
| [0, 3] | Accuracy | 20.4 | 71.6 | 75.4 | 84.6 | 85.2 |
|        | FID | 109.1 | 67.6 | 47.4 | 31.3 | 22.4 |

Table 7: Effect of stroke width range on accuracy and FID.

## F.6 NUMBER OF SAMPLES GENERATED FROM THE SIMULATOR.

We fixed the number of images to 60,000 across all experiments and iterations, following the DPImageBench evaluation protocol (Gong et al., 2025) for fair and consistent comparison across methods. Additionally, we run MNIST experiments ($\varepsilon = 1$) with 30,000 and 120,000 samples in Tab. 8.

Consistent with the claims in the original PE paper (Appendix N in Lin et al. (2023)), we see that more samples don't always improve results. On one hand, more samples increase the chance of generating examples close to the private data, potentially improving synthetic data. On the other hand, a larger sample size flattens the DP Nearest Neighbors Histogram, reducing its signal-to-noise ratio and possibly degrading quality. The overall effect is nuanced. Following Lin et al. (2023), we recommend setting the number of generated samples to be close to the number of private samples.

| #Samples | FID | Accuracy |
|----------|-----|----------|
| 30,000 | 14.4 | 87.2 |
| 60,000 | 20.7 | 89.1 |
| 120,000 | 32.9 | 85.4 |

Table 8: Performance comparison with different numbers of samples.

## F.7 THE IMPACT OF DISTRIBUTION ALIGNMENT

On CelebA, we show that Sim-PE with a simulator that aligns well with the private data outperforms standard PE (§ 4.3 and Tab. 1). However, Sim-PE with a simulator less aligned with the private data performs worse than PE (§ 4.4 and Tab. 2).

To provide a more controlled experiment, we fix the simulator type and only vary the alignment between the simulator and private data.

**Experiment settings.** We use the same 1.2M simulator-generated images from a computer graphics renderer as in § 4.3 and artificially adjust their alignment to CelebA. Specifically, for each image, we compute its distance to the closest CelebA image in the Inception embedding space, sort all images by this distance, and divide them into five subsets, $D_0, ..., D_4$ (e.g., the closest 0.24M images form $D_0$). Fig. 11 confirms that the FID between $D_i$ and CelebA increases with $i$, meaning that $D_0$ is the most aligned and $D_4$ the least. We then apply Sim-PE to each $D_i$ independently.

**Results.** Fig. 12 shows that as alignment decreases, the sample quality generally drops. Specifically, Sim-PE with $D_0, ..., D_3$ yields better classification accuracy than PE, while $D_4$ yields worse results than PE. This confirms that Sim-PE's performance is influenced by the degree of alignment. Additionally, in all cases, Sim-PE's FID is better than the original simulator data's FID, demonstrating Sim-PE's ability to select useful samples.

## F.8 CLASS LABEL INFORMATION FROM THE SIMULATORS

For simulator 1, the target class label (i.e., the digit) is fully controlled by one parameter. For simulators 2 and 3, the target class label (i.e., the gender) is not directly controlled by any parameter, but could potentially be obtained by an external image gender classifier. One benefit of using domain-specific simulators is that we can potentially use the class label information to enhance

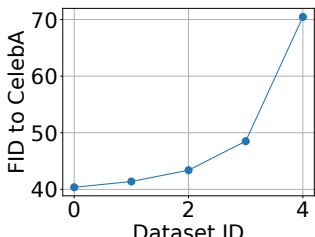

Figure 11: FID between the 5 constructed subsets of DIGIFACE-1M and CelebA.

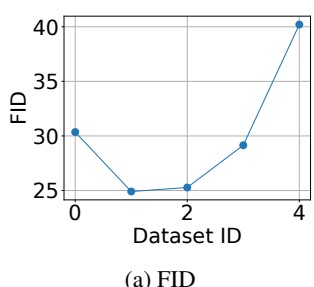

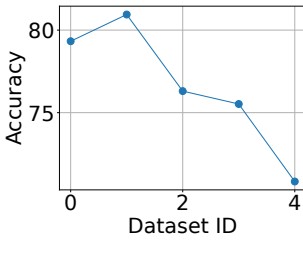

(a) FID

(b) Classification accuracy

Figure 12: Results of Sim-PE-generated data using the subsets in Fig. 11 ($\epsilon = 10$).

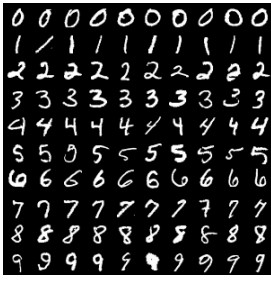

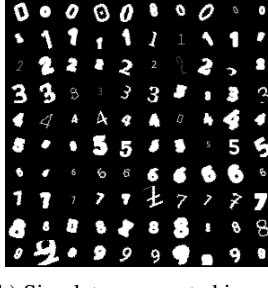

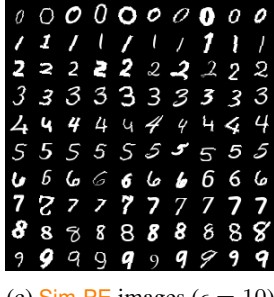

(a) Real (private) images

(b) Simulator-generated images

(c) Sim-PE images ($\epsilon = 10$)

Figure 13: The real and generated images on MNIST under the "**ClassAvail**" setting. Each row corresponds to one class. The simulator generates images that are very different from the real ones. Starting from these bad images, Sim-PE can effectively guide the generation of the simulator towards high-quality images that are more similar to real data.

data quality (Wood et al., 2021; Bae et al., 2023). To get a more comprehensive understanding of Sim-PE, we consider two settings: **(1) Class label information is unavailable (abbreviated as "ClassUnavail").** We artificially make the problem more challenging by assuming that the class label information is *not* available. Therefore, Sim-PE has to learn to synthesize images with the correct class by itself. Our main experiments are based on this setting. **(2) Class label information is available (abbreviated as "ClassAvail").** On MNIST, we further test how Sim-PE can be improved if the class label information is available. In this case, the RANDOM_API and VARIATION_API (Eqs. (1) and (2)) are restricted to draw parameters from the corresponding class (i.e., the digit is set to the target class).

**Results: Class label information from the simulators can be helpful.** The results are presented in Tab. 9 and Fig. 13. We observe that with digit information, the simulator-generated data achieves much higher classification accuracy (92.2%), although the FID remains low due to the generated digits exhibiting incorrect characteristics (Fig. 13b). The fact that Sim-PE outperforms the simulator in both FID and classification accuracy across all settings suggests that Sim-PE effectively incorporates private data information to enhance both data fidelity and utility, even when compared to such a strong baseline. As expected, Sim-PE under **ClassAvail** matches or surpasses the results obtained in **ClassUnavail** across all settings, suggesting the usefulness of leveraging class label information.

Table 9: Accuracy (%) of classifiers trained on synthetic images and FID of synthetic images on MNIST under the "**ClassAvail**" setting. See Tab. 1 for results under the "**ClassUnavail**" setting for reference.

| Algorithm | FID ↓ | | Classification Acc. ↑ | |
|---|---|---|---|---|
| | $\epsilon = 1$ | $\epsilon = 10$ | $\epsilon = 1$ | $\epsilon = 10$ |
| Simulator | 86.0 ($\epsilon = 0$) | | 92.2 ($\epsilon = 0$) | |
| Sim-PE | **20.7** | **8.6** | **93.9** | **95.5** |

## G  EXPERIMENTAL DETAILS

In this section, we provide more experimental details.

### G.1  MNIST WITH TEXT RENDERING PROGRAM

**The categorical parameters include:** (1) Font. We use Google Fonts (Google, 2022), which offers 3589 fonts in total. (2) Text. The text consists of digits '0' - '9'. **The numerical parameters include:** (1) Font size, ranging from 10 to 29. (2) Stroke width, ranging from 0 to 2. (3) Digit rotation degree, ranging from $-30°$ to $30°$.

Tabs. 10 and 11 show their variation degrees. The total number of PE iterations is 4. Following Gong et al. (2025), we set the number of generated samples to be 60,000.

Table 10: The configurations of the categorical parameters in MNIST with Text Rendering Program experiments.

| Categorical Parameter ($\xi$) | Feasible Set ($\Xi$) | Variation Degrees ($\beta$) across PE Iterations |
|---|---|---|
| Font | 1 - 3589 | 0.8, 0.4, 0.2, 0.0 |
| Text | '0' - '9' | 0, 0, 0, 0 |

Table 11: The configurations of the numerical parameters in MNIST with Text Rendering Program experiments.

| Numerical Parameter ($\phi$) | Feasible Set ($\Phi$) | Variation Degrees ($\alpha$) across PE Iterations |
|---|---|---|
| Font size | [10, 30] | 5, 4, 3, 2 |
| Font rotation | [-30, 30] | 9, 7, 5, 3 |
| Stroke width | [0, 2] | 1, 1, 0, 0 |

### G.2  CELEBA WITH GENERATED IMAGES FROM COMPUTER GRAPHICS-BASED RENDER

The variation degrees $\gamma$ across PE iterations are [1000, 500, 200, 100, 50, 20]. The total number of PE iterations is 6. Following Gong et al. (2025), we set the number of generated samples to be 60,000.

### G.3  CELEBA WITH RULE-BASED AVATAR GENERATOR

The full list of the categorical parameters are

- Style
- Background color
- Top
- Hat color
- Eyebrows
- Eyes
- Nose
- Mouth

- Facial hair
- Skin color
- Hair color
- Facial hair color
- Accessory
- Clothing
- Clothing color
- Shirt graphic

These are taken from the input parameters to the library (Escartín, 2021). There is no numerical parameter.

For the experiments with only the simulator, the variation degrees $\beta$ across PE iterations are [0.8, 0.6, 0.4, 0.2, 0.1, 0.08, 0.06]. The total number of PE iterations is 7. Following Gong et al. (2025), we set the number of generated samples to be 60,000.

For the experiments with both foundation models and the simulator, we use a total of 5 PE iterations so as to be consistent with the setting in Gong et al. (2025). For the RANDOM_API and the first PE iteration, we use the simulator ($\beta = 0.8$). For the next 4 PE iterations, we use the same foundation model as in Lin et al. (2023) with variation degrees [96, 94, 92, 90]. Following Gong et al. (2025), we set the number of generated samples to be 60,000.

## H   EFFICIENCY EVALUATION

As simulators could be much cheaper to generate samples than the foundation models, we show in this section that *Sim-PE is much more efficient than PE* in our experiments.

Note that the only difference between Sim-PE and PE is the RANDOM_API and VARIATION_API. Therefore, we focus on comparing the computation time and peak CPU/GPU memory of the APIs. Tabs. 12 to 14 show that Sim-PE APIs require far less time than PE. For instance, on MNIST, each PE's API call takes over *2400 GPU seconds*, whereas Sim-PE takes less than *30 CPU seconds*– an 80x speedup, not to mention the lower cost of CPU than GPU. Consequently, each Sim-PE iteration is much faster than PE (Tab. 15). The only Sim-PE operation requiring GPU is computing the embedding and nearest neighbors of simulator-generated data (§ 3.3), which is a one-time cost per dataset. Even this one-time process is significantly faster than one PE API call. Sim-PE with data access requires more CPU memory to store simulator-generated data, but this can be easily optimized by loading only the needed data.

|  |  | Time | Peak CPU Memory | Peak GPU Memory |
|---|---|---|---|---|
| **PE** | RANDOM_API | 3920.30 seconds (GPU) | 15292.98 MB | 13859.10 MB |
|  | VARIATION_API | 2422.44 seconds (GPU) | 15880.25 MB | 16203.89 MB |
| **Sim-PE** | RANDOM_API | 27.17 seconds (CPU) | 758.90 MB | 0 MB |
|  | VARIATION_API | 18.51 seconds (CPU) | 1083.11 MB | 0 MB |

Table 12: Efficiency comparison of PE and Sim-PE on the MNIST dataset. Tested on a Linux server with `AMD EPYC 7V12 64-Core Processor` and one `NVIDIA RTX A6000 GPU`.

|  |  | Time | Peak CPU Memory | Peak GPU Memory |
|---|---|---|---|---|
| **PE** | RANDOM_API | 39272.33 seconds (GPU) | 15293.29 MB | 13859.10 MB |
|  | VARIATION_API | 31028.24 seconds (GPU) | 15879.47 MB | 16203.89 MB |
| **Sim-PE** | RANDOM_API | 0.04 seconds (CPU) | 61416.62 MB | 0 MB |
|  | VARIATION_API | 0.03 seconds (CPU) | 61427.29 MB | 0 MB |
| **Sim-PE Setup** |  | 1450.94 seconds (GPU) | 61420.69 MB | 3728.16 MB |

Table 13: Efficiency comparison of PE and Sim-PE (with data access) on the CelebA dataset. Tested on a Linux server with `AMD EPYC 7V12 64-Core Processor` and one `NVIDIA RTX A6000 GPU`.

| | | Time | Peak Memory | Peak GPU Memory |
|---|---|---|---|---|
| **PE** | RANDOM_API | 39272.33 seconds (GPU) | 15293.29 MB | 13859.10 MB |
| | VARIATION_API | 31028.24 seconds (GPU) | 15879.47 MB | 16203.89 MB |
| **Sim-PE** | RANDOM_API | 46.69 seconds (CPU) | 811.72 MB | 0 MB |
| | VARIATION_API | 48.78 seconds (CPU) | 1067.93 MB | 0 MB |

Table 14: Efficiency comparison of PE and Sim-PE (with code access) on the CelebA dataset. Tested on a Linux server with `AMD EPYC 7V12 64-Core Processor` and one `NVIDIA RTX A6000 GPU`.

| | MNIST | CelebA (Sim-PE with data access) | CelebA (Sim-PE with code access) |
|---|---|---|---|
| **PE** | 22243.32 | 279695.52 | 279695.52 |
| **Sim-PE** | 607.95 | 441.63 | 880.38 |

Table 15: The runtime (seconds) of one iteration in PE and Sim-PE. Tested on a Linux server with `AMD EPYC 7V12 64-Core Processor` and one `NVIDIA RTX A6000 GPU`.

# I   EXTENDED DISCUSSIONS

## I.1   THE PREVALENCE AND IMPORTANCE OF SIMULATORS

In this paper, we define *simulators* as any data synthesizers that do not rely on neural networks. These simulators are widely used across various applications due to their unique advantages over neural network-based approaches.

**The prevalence of simulators.** Despite the widespread adoption of foundation models, simulators remain highly prevalent. Here are a few notable examples:

- **Genesis:**[4] A physics-based simulator used in robotics, embodied AI, and physical AI applications. Since its release in late 2024, it has received over 24k GitHub stars and 84k downloads as of April 2024.
- **Blender:**[5] A rendering framework widely used for image and video production, including in movie-making (see examples at Blender Studio[6]). One simulator used in our face experiment (the DIGIFACE-1M dataset) is built on Blender.
- **Unreal:**[7] A widely adopted game engine with image/video rendering capability. It holds a 14.85% market share in the game development industry.[8] Note that competing engines also qualify as "simulators" under our definition.

**The importance of simulators.** As demonstrated above, simulators continue to play a crucial role in industry applications. Even synthetic data generation also frequently relies on simulators rather than foundation models (e.g., META-SIM (Kar et al., 2019), DIGIFACE-1M (Bae et al., 2023)). This preference stems from several unique advantages of simulators over foundation models:

- **Rich annotations**: Simulators provide additional structured labels due to explicit control over the data generation process. For instance, besides face images, FACE SYNTHETICS (Wood et al., 2021) offers pixel-level segmentation masks (e.g., identifying eyes, noses), which are highly valuable for downstream tasks such as face parsing.
- **Task-specific strengths**: Certain generation tasks remain challenging for foundation models. For example, despite recent advances, foundation models still struggle with generating images containing text (Rombach et al., 2022), whereas simulators can handle it easily. Our MNIST experiment was specifically designed to highlight this distinction.
- **Domain-specific strengths**: In domains like networking, where robust foundation models are lacking, network simulators such as ns-3[9] remain a more reliable and scalable solution.

---

[4] https://genesis-embodied-ai.github.io/
[5] https://github.com/blender/blender
[6] https://studio.blender.org/films/
[7] https://www.unrealengine.com/en-US
[8] https://6sense.com/tech/game-development/unreal-engine-market-share
[9] https://www.nsnam.org/

- **Greater reliability and control**: Because simulators model the data generation process, they mitigate issues such as generating anatomically incorrect images (e.g., faces without noses[10]).

That said, there are indeed data domains for which no suitable simulators exist. However, the goal and contribution of this work is not to cover every possible data domain–which is inherently infeasible–but rather to significantly expand the range of domains where PE can be applied. For example, we demonstrate its effectiveness on challenging domains for PE such as text rendering and face generation.

Moreover, the proposed approach is not limited to using simulators. The method in § 3.3 can also be applied to any public dataset, as demonstrated in App. D. This highlights the broader potential of our approach that sidesteps the limitation of requiring a suitable simulator.

## I.2 DISCUSSION ON THE CHOSEN METRICS

In the main experiments, we use FID and the accuracy of downstream classifiers as the metrics.

**Why we pick these metrics.** We chose FID and classification accuracy because they are the most widely used metrics in DP image synthesis. *Most of the baselines we compared to utilize only these two metrics or just one of them (Gong et al., 2025).* The only exception is GS-WGAN, which also uses Inception Score (IS) (Salimans et al., 2016). However, IS is more relevant for natural images (e.g., ImageNet images) because it evaluates the diversity across ImageNet classes, making it unsuitable for the datasets we considered in the main paper. Additionally, for the downstream task (ten-class classification for MNIST and binary classification for CelebA), we also follow prior works (Gong et al., 2025).

**The differences between these metrics.** FID measures the "fidelity" of synthetic data by mapping both synthetic and private data to an embedding space, approximating each with a Gaussian distribution, and calculating the Wasserstein distance between the two distributions. Classification accuracy measures the "utility" of synthetic data by evaluating its performance when used to train a downstream classifier, simulating real-world use where users expect the synthetic data to support good classifier performance on real data. *FID and classification accuracy are complementary*: FID focuses on distribution-level closeness, while classification accuracy is sensitive to outliers or samples near the classification boundary. A high score in one does not necessarily correlate with a high score in the other. This is why we often see cases where the best methods for these metrics differ (Gong et al., 2025).

## I.3 BIAS IN SIMULATORS

Because Sim-PE relies on simulators, biases inherent in the simulator can be transferred to the synthetic data produced by Sim-PE.

## I.4 THE ORDERING OF APPLYING THE SIMULATOR AND THE FOUNDATION MODEL

In § 3.4, we propose to use simulators in early iterations and then switch to foundation models. The reverse ordering–"foundation model → simulator"–is also feasible with appropriate modifications. In this setup, when transitioning from the foundation model to the simulator, one would need to find the optimal simulator parameters corresponding to each sample generated by the foundation model, potentially using gradient-free optimization methods.

However, we only explore "simulator → foundation model" in the paper for two main reasons:

- This ordering keeps the approach simple and avoids the need for special treatments like the one described above.
- It is also more promising in data quality. Simulators can generate diverse samples that span a wide range of variations, making them ideal for initialization. However, their image quality is bad. Foundation models, on the other hand, are better at producing high-fidelity, realistic samples, which is more desirable in the final stages of the pipeline, which creates the final synthetic data.

---

[10]https://www.reddit.com/r/StableDiffusion/comments/1e7dd62/weird_and_distorted_images_with_a1111_sd3/

## J    THE USE OF LARGE LANGUAGE MODELS (LLMS)

The paper was mostly written by the authors, with LLMs assisting in grammar correction and language polishing.

