# OpenReview forum: "Differentially Private Synthetic Data via APIs 3: Using Simulators Instead of Foundation Models"
_ICLR.cc/2026/Conference — Submitted to ICLR 2026_

### Official Review · Reviewer_kg1H · 2025-10-16

**Soundness:** 2
**Presentation:** 2
**Contribution:** 2
**Rating:** 2
**Confidence:** 5

**Summary:**

This paper proposes Sim-PE, an extension of Private Evolution (PE) that replaces foundation models with simulators as the generative backend. Sim-PE supports two scenarios: when simulator APIs are accessible, the RANDOM and VARIATION APIs are implemented by sampling from the feasible parameter space; and when only a large pool of simulator-generated data is available, the VARIATION API is realized through nearest-neighbor sampling. Experiments on image datasets show that Sim-PE outperforms PE under strong distribution shifts and further benefits from combining simulators with foundation models.

**Strengths:**

- A simple but effective approach that 1) improves over PE; 2) demonstrates the generality of the core mechanism of PE

- Comprehensive experiments on image synthesis, covering diverse simulator types, access settings, and combinations with foundation models

**Weaknesses:**

- The technical novelty is limited. The PE framework remains essentially unchanged, and adapting its RANDOM and VARIATION APIs to simulators is straightforward. Key limitations of PE persist, including but not limited to mode collapse, sampling bias (e.g., top-K), and non-monotonic performance w.r.t. iterations.

- The main contribution lies in identifying simulators as a new data source rather than introducing a new algorithmic insight. The empirical study is confined to image data, where the use of simulators is somewhat obvious and less compelling. More interesting domains such as robotics or physical simulation are left unexplored.

- The strong results (Sec 4.2) largely stem from cases where the simulator distribution is already well aligned with the private data (rendering digits with Python PIL). It remains unclear how to choose or adapt a simulator for arbitrary private datasets, which limits the general applicability of the proposed approach. As a concrete question, what simulator would you consider if the goal is to generate clinical or biometric data?

Overall, this work identifies an interesting direction but lacks sufficient technical depth for ICLR.

**Questions:**

Sim-PE heavily depends on the choice of the similarity metric or embedding space used for the DP-NN voting and for the VARIATION API (nearest-neighbor sampling). In your experiments, this embedding comes from pretrained vision models like CLIP, which themselves are trained on massive web data. Doesn’t this reintroduce reliance on large foundation models, the very dependency Sim-PE claims to eliminate? How would Sim-PE operate in a domain where no strong pretrained embedding exists?

---

> ### Author Response · Authors · 2025-11-13
> **Rebuttal (1/3)**
>
> *Thank you very much for your insightful comments. We have conducted all the suggested experiments, which were very helpful. We provide responses to each of your points below.*
>
> ----
>
> ----
>
> > ### Q1. novelty and contribution
>
> **We respectfully but strongly disagree with the reviewer’s assessment of our contribution.** Below, we clarify our perspective on Sim-PE and, more broadly, on the trajectory of DP synthetic data research.
>
> **1. Unreasonable expectation for a single paper.**  We agree that PE has *orthogonal* limitations that are worthwhile to address. However, for any given paper, we can always provide a long list of *orthogonal* problems that it does not solve—no paper can realistically address all open challenges, and a “perfect’’ method without any limitation does not even exist. Not solving these orthogonal limitations should not constitute grounds for rejection. Using the same evaluation standards applied to prior publications in this area, we believe our paper provides a clear and sufficient contribution, with both novelty and potential impact, as detailed below.
>
> **2. Using simulators for DP synthetic data has never been explored before.**  The central idea—that *PE can operate effectively with simulators*—is both non-obvious and fundamentally new. Among the 60+ follow-up works citing PE (per Google Scholar), *none* investigate non–foundation-model data synthesizers such as simulators. More broadly, throughout the entire DP synthetic data literature, the simulators we explored have never been used as a DP data generator. Sim-PE therefore not only broadens the applicability of PE, but also introduces a new paradigm that unlocks the potential of simulators for DP data generation. We believe this reframes an important part of the design space for future work in DP synthetic data.
>
> **3. Simplicity is a strength, not a weakness, and this seemingly simple approach is non-trivial to devise.**  We strongly disagree with the implication that a “simple approach’’ is inherently non-novel or insufficient for acceptance. As detailed in Section 3.3 and empirically validated in Appendix F.1, the most straightforward strategies fail for this problem; arriving at this “simple” solution required careful reasoning and substantial insight. The fact that such an easy-to-implement idea was missed by all prior work highlights its conceptual novelty rather than detracting from it.
> Moreover, simplicity is a practical strength that makes methods easier to use, deploy, and build upon, rather than a weakness.
>
> **4. Fairness and consistency in evaluating contributions.**  Before Private Evolution (PE), DP fine-tuning was the dominant paradigm for DP synthetic data generation. Most works in this line rely on *the same DP mechanism*, DP-SGD (proposed in 2016), and many famous and important works differentiate themselves primarily by *changing the models*. Representative examples include:
>
>     - *Differentially Private Fine-Tuning of Language Models* (Yu et al., ICLR 2022, >500 citations): DP-SGD on RoBERTa and GPT2.
>     - *Synthetic Text Generation with Differential Privacy: A Simple and Practical Recipe* (Yue et al., ACL 2023 Honorable Mention, >100 citations): DP-SGD on GPT2.
>     - *Differentially Private Diffusion Models* (Tim et al., TMLR 2023, >150 citations): DP-SGD on diffusion models.
>     - PrivImage (Li et al., USENIX Security 2024): DP-SGD on diffusion models.
>     - *Differentially Private Latent Diffusion Models* (Liu et al., TMLR 2024): DP-SGD on latent diffusion models.
>
> Conceptually, PE plays the same role as DP-SGD—a *DP mechanism* for synthetic data generation. Under this analogy, *applying DP-SGD on new models is directly parallel to applying PE on new APIs*. Since applying DP-SGD on new models is widely accepted and regarded as a sufficient contribution, we argue that our work should be treated fairly in the same way.
>
> **Summary.**  Across all dimensions—fairness in evaluating contributions relative to prior work, the non-obviousness of the proposed approach, the practical strength of the method’s simplicity, the conceptual novelty of enabling PE to operate with simulators, and broader impact of introducing simulator-based generation into the DP landscape—we believe the contribution of Sim-PE is both solid and significant. We also hope that **our new experiments demonstrating Sim-PE’s applicability in a completely different domain (networking)** help reassure the reviewer of the value and potential of the approach (see Q2).
>
> We thank the reviewer for sharing their perspective and welcome further discussion.

---

> > ### Author Response · Authors · 2025-11-13
> > **Rebuttal (2/3)**
> >
> > > ### Q2.  The empirical study is confined to image data. More interesting domains such as robotics or physical simulation are left unexplored.
> >
> > Thank you for the suggestion. We conducted an additional experiment on networking data. We selected the networking domain for two reasons:
> > (1) There are currently no good foundation models for networking data, making DP synthetic data generation for this domain both more challenging and more impactful;
> > (2) There are no strong pretrained embedding models in this domain, making it an ideal setting to address your Q4 (“How would Sim-PE operate in a domain where no strong pretrained embedding exists?”).
> >
> > We follow the experimental setup of NetDPSyn [1], a recent paper published at IMC 2024, one of the top networking conferences.
> >
> > **Data.**  We aim to generate differentially private *network flow header* data using the TON dataset [1] as the private dataset. Each sample contains the following fields:
> > - Source IP address
> > - Destination IP address
> > - Source port
> > - Destination port
> > - Protocol (e.g., UDP, TCP)
> > - Timestamp of the first packet
> > - Duration of the flow
> > - Number of packets
> > - Number of bytes
> > - Flow label (10 classes, e.g., normal, background, DDoS)
> >
> > **Simulator.**  We use a very simple simulator: the `RANDOM_API` generates random flows, and the `VARIATION_API` perturbs individual fields of a flow (e.g., modifying the IP address).
> >
> > **Embedding.**  To keep the setup minimal, we embed samples directly in their normalized raw format (one-hot encodings for categorical fields and normalized numeric values for continuous ones). Thus, **the embedding does not rely on any machine learning model**.
> >
> > **Metrics.**  Following [1], we report:
> > - **1-way and 2-way Total Variation Distance (TVD)** between synthetic and private data (lower is better)
> > - **Classification accuracy**, obtained by training a classifier on synthetic data and testing on held-out private data (higher is better). This represents the downstream task of network flow data.
> >
> > **Privacy budget.**  As in [1], we set ε = 2.
> >
> > **Results.**  The table below shows how the metrics evolve across Sim-PE iterations. Key observations:
> >
> > 1. **All metrics steadily improve across iterations.** Final performance is significantly better than the initial random samples (e.g., accuracy improves by **16.76%**). This demonstrates that Sim-PE can work effectively for networking data *even without* ML-based embeddings or a sophisticated simulator.
> >
> > 2. **Compared to Figure 3 in [1], this simple approach already outperforms two baselines in accuracy.** Notably, NetShare—a widely used DP fine-tuning–based synthesizer in networking (130+ citations)—is surpassed by Sim-PE in classification accuracy. This highlights the strong potential of Sim-PE beyond the image domain.
> >
> > | Iteration | Accuracy | 1-way TVD | 2-way TVD |
> > |-----------|----------|-----------|-----------|
> > | 0         | 56.48    | 0.29      | 0.46      |
> > | 1         | 56.48    | 0.29      | 0.45      |
> > | 2         | 56.48    | 0.28      | 0.44      |
> > | 3         | 56.75    | 0.27      | 0.43      |
> > | 4         | 57.08    | 0.27      | 0.42      |
> > | 5         | 58.28    | 0.26      | 0.41      |
> > | 6         | 60.29    | 0.25      | 0.40      |
> > | 7         | 61.28    | 0.24      | 0.38      |
> > | 8         | 63.47    | 0.23      | 0.37      |
> > | 9         | 65.95    | 0.22      | 0.36      |
> > | 10        | 67.37    | 0.21      | 0.35      |
> > | 11        | 68.47    | 0.21      | 0.33      |
> > | 12        | 72.18    | 0.20      | 0.32      |
> > | 13        | 72.77    | 0.19      | 0.31      |
> > | 14        | 73.24    | 0.18      | 0.30      |
> >
> > [1] https://arxiv.org/abs/2409.05249

---

> ### Author Response · Authors · 2025-11-13
> **Rebuttal (3/3)**
>
> > ### Q3. It remains unclear how to choose or adapt a simulator for arbitrary private datasets. what simulator would you consider if the goal is to generate clinical or biometric data?
>
> * We agree with the reviewer that Sim-PE cannot be applied to *arbitrary* private datasets. **However, DP fine-tuning—the dominant approach in DP synthetic data—also has *not* been successfully applied to arbitrary
>  domains.** In fact, it is unclear whether applicability to “arbitrary private datasets’’ is even achievable for any method. Therefore, we do not believe this constitutes a reasonable basis for rejection. Given the diverse domains where Sim-PE already works (as demonstrated in both the paper and the new experiments in the rebuttal), we believe the contribution is already substantial.
> * For some clinical or biometric data, relevant foundation models already exist (see examples below). In such cases, we believe *PE with foundation models* is more promising. We emphasize that the goal of Sim-PE is *not* to replace PE with foundation models, but to complement it; when suitable foundation models are available, users should simply apply PE with those models instead of using Sim-PE.
>     - GPT for medical record (https://arxiv.org/abs/2402.04400)
>     - Diffusion models for images of histological lymph node sections (https://arxiv.org/pdf/2305.15560)
>     - MAMBO for mammograms (https://arxiv.org/abs/2506.08677)
>     - DSB-GAN for biometric data (https://www.sciencedirect.com/science/article/abs/pii/S0141938222000865)
>
> > ### Q4. This embedding comes from pretrained vision models like CLIP, which themselves are trained on massive web data. How would Sim-PE operate in a domain where no strong pretrained embedding exists?
>
> Thank you for the great question.
> * **First, we would like to clarify that we did *not* use CLIP as the embedding model** (although it also works). Instead, we used the Inception embedding pretrained on ImageNet. Inception is an older and relatively weak model released in 2015.
> * **Second, Sim-PE can also work *without* any pretrained embedding model.** As shown in Q2, using raw samples directly (with simple normalization and no learned embedding) is sufficient for Sim-PE to perform well on networking data. This demonstrates that Sim-PE does not rely on strong pretrained embeddings and might still function effectively when such models are unavailable.
>
>
> ----
>
> ----
>
> *Finally, we would like to express our sincere appreciation for your thorough evaluation and insightful suggestions. We are grateful for the time you dedicated to improving our paper.*

---

> > ### Comment · Reviewer_kg1H · 2025-11-19
> >
> > I appreciate the authors’ rebuttal and additional experiment. However, the core concerns raised in my initial review are not resolved. In fact, the rebuttal raises additional issues regarding the positioning and scholarly substance of the work.
> >
> > - Much of the rebuttal focuses on how papers “should be evaluated”, what constitutes “fairness”, and why certain expectations are “unreasonable”. These arguments do not address the scientific questions raised. **A submission must be evaluated on its own technical and empirical contributions, not on comparisons to prior publication standards or claims about fairness.**
> >
> > - Statements such as *none of the 60+ PE follow-ups used simulators* do not imply importance or depth; many unexplored directions remain unexplored simply because they are not interesting or add little. **The claim that prior DP-SGD works *primarily changed models* is a very inaccurate and scientifically irresponsible characterization of the literature**; many cited works introduce substantial technical insights (e.g., PEFT and scaling behavior in Yu et al.) beyond changing the base model. Downweighting prior work does not help strengthen this paper at all.
> >
> > - I see two ways for papers in this area to make a meaningful contribution:
> >   - **Non-trivial algorithmic or theoretical innovation**. I believe the authors would agree that Sim-PE does not offer this.
> >   - **Enabling new applications not achievable by existing methods**. Unfortunately, this is also not demonstrated even with the new experiments included. The original submission focuses solely on image data. The added networking experiment in the rebuttal is in fact standard **tabular data**. DP tabular data synthesis is a very mature field with well-established toolkits and benchmarks. The cited reference (which is highlighted as a publication at “top networking conferences”) does not even compare against AIM, a state-of-the-art method in this area. I don’t see what PE-based methods offer beyond existing DP tabular synthesizers.
> >   - Without meaningful contributions in any of the above ways, the authors’ emphasis on conceptual contributions is not well-grounded.
> >
> > - As a summary, the rebuttal does not resolve the fundamental concerns:
> >   - No substantial technical novelty
> >   - No genuinely new or difficult application domain
> >   - Overstated claims of conceptual impact and reliance on argumentative framing rather than scientific evidence
> >
> > For these reasons, the submission remains below the bar for ICLR. My score and recommendation remain unchanged.
> >
> > I also note that the additional results in the rebuttal are collected within roughly one to two days. While I appreciate the authors’ responsiveness and am willing to take this positively as an indication that they took reviewers’ feedback seriously and that their method is easy to implement, such a short timeline generally does not allow for careful experimental design, selection of appropriate baselines, thorough evaluation or rigorous analysis (unless the results were prepared in advance).

---

> > > ### Author Response · Authors · 2025-11-19
> > >
> > > We sincerely thank the reviewer for their willingness to engage in thoughtful discussion on these important perspectives, and we fully respect the reviewer’s final decision.
> > > **However, we would like to clarify a few key misunderstandings regarding our statements and intentions.**
> > >
> > > ---
> > >
> > > ### **Paper evaluation criteria**
> > > We completely agree that *“A submission must be evaluated on its own technical and empirical contributions”*. However, our main disagreement centers on **novelty and contribution**, which is inherently **subjective**. As different researchers come with different backgrounds and perspectives, having different opinions on *what is considered as novel/contribution* and *how much novelty/contribution is sufficient* is natural.
> > > Our intention was therefore to **calibrate our expectations** using influential past papers. We believe this is a fair and reasonable way to evaluate papers, and under this framing, we felt our contribution was sufficient.
> > >
> > > ---
> > >
> > > ### **Works on DP-SGD**
> > > We never claimed that *“prior DP-SGD works primarily changed models”*, which would indeed be an incorrect claim.  Our original statement (quoted verbatim) is:
> > >
> > > > *“Most works in this line rely on the same DP mechanism, DP-SGD (proposed in 2016), and many famous and important works differentiate themselves primarily by changing the models.” (citations listed in the rebuttal)*
> > >
> > > Specifically, our claims are two-fold:
> > > (1) most works in DP fine-tuning rely on the **same DP mechanism (DP-SGD)**; and
> > > (2) **many famous works** primarily differentiate themselves through model changes—as supported by the citations we provided.
> > >
> > > We believe our original sentence stated these two points clearly. The reviewer’s rephrasing appears to conflate them by taking *“most works”* from the first claim and *“differentiate themselves primarily by changing the models”* from the second claim, thereby creating an interpretation we did not make and that is factually incorrect.
> > >
> > > ---
> > >
> > > ### **Contributions**
> > >
> > > **“Non-trivial algorithmic or theoretical innovation”**: As explained in our rebuttal, correctly using the simulator is non-trivial:
> > >
> > > > *“As detailed in Section 3.3 and empirically validated in Appendix F.1, the most straightforward strategies fail for this problem; arriving at this “simple” solution required careful reasoning and substantial insight.”*
> > >
> > > **“Enabling new applications not achievable by existing methods”**: While generating DP networking data with PE has not previously been demonstrated using PE-based methods, we fully agree that other approaches (including AIM) are applicable (we also discussed another baseline, NetShare, in our rebuttal).
> > > We appreciate the reviewer highlighting AIM. Although the “missing baseline” comment pertains to the reference we followed, we will certainly incorporate this suggestion when conducting the full experiment.
> > >
> > > ---
> > >
> > > ### **Completing rebuttal experiments in two days**
> > > While we welcome the discussions on how to strengthen these experiments, using the subjective judgment that *“such a short timeline generally does not allow for careful experimental design, selection of appropriate baselines, thorough evaluation or rigorous analysis ”* to imply that our experiments lack rigor is neither a scientific approach nor irrespective of the substantial effort we invested during those two days.
> > >
> > > ---
> > >
> > > ### **Summary**
> > > In summary, we believe our only remaining disagreement appears to be whether the **novelty and contribution** are sufficient for ICLR. While we believe they are, we genuinely appreciate the reviewer’s constructive suggestions—including more algorithmic extensions and additional application domains—which will indeed help further strengthen the paper. We will take them seriously moving forward.

---

> ### Comment · Reviewer_kg1H · 2025-11-19
> **Final comment**
>
> I appreciate the authors’ follow-up response. However, I would like to clarify several points and close the discussion.
>
> **On the characterization of prior DP-SGD work.** The authors’ original statement in the rebuttal, *“many famous and important works differentiate themselves primarily by changing the models”*, is exactly what I was responding to. Regardless of how the authors parse my comment, the implication was clear in context. I strongly disagree with this characterization of the literature. Many of the cited papers introduce technical insights that go far beyond model changes. More importantly, this line of argument is not relevant to evaluating the current submission. Whether prior papers did or did not “differentiate themselves primarily by changing models” has no bearing on the scientific merit of this work.
>
> **On the timing of the new experiments.** My comment regarding the short turnaround time was not intended to accuse the authors of lacking rigor. If it offended the authors, I apologize. It simply reflects a view about experimental practice that I personally hold: meaningful experimental design, proper baseline selection, and thorough evaluation typically require time. And to be fair, I do think there is large room of improvement in terms of baseline selection and experimental design.
>
> **On what would constitute a compelling contribution.** Ultimately, my assessment is about the scientific contribution of Sim-PE. I would genuinely like to see settings where Sim-PE provides value beyond existing approaches. However, applying it to image or tabular data where there exist strong, specialized DP synthesis methods does not convincingly demonstrate this. The potential usefulness of Sim-PE likely lies in domains where simulators provide structure and inductive biases unavailable to other methods. Those scenarios were not explored in the submission or rebuttal.

---

> > ### Author Response · Authors · 2025-11-19
> >
> > Thank you for the clarifications and suggestions. We appreciate it.

---

> > > ### Author Response · Authors · 2025-11-21
> > > **Update**
> > >
> > > Dear Reviewer kg1H,
> > >
> > > Thank you again for your helpful suggestions and active discussions. In the past day, we have been carefully considering your comment about demonstrating the value of Sim-PE in domains where *"simulators provide structure and inductive biases unavailable to other methods."* We are planning to explore an additional experimental setting aligned with this goal. If we are able to complete the experiments before the end of the discussion period, we will update them here; otherwise, we will include the results in the revision.
> > >
> > > Sincerely,
> > >
> > > *The Authors*

---

> > > > ### Comment · Reviewer_kg1H · 2025-11-21
> > > >
> > > > Thank you for taking my comment seriously and for considering additional experiments in domains where simulator-based methods could offer unique advantages. I appreciate the effort. I’d like to set expectations upfront: given the tight rebuttal timeline and the holiday season at year’s end, I may not be able to follow up on new results in a timely manner. Nevertheless, I’m glad to see the exploration moving in a direction that could meaningfully strengthen the paper, and I hope these investigations prove useful for your revision.

---

> ### Author Response · Authors · 2025-11-25
> **Additional Experiments in Domains where Sim-PE could Offer Unique Advantages (1/2)**
>
> **Dear Reviewer kg1H,**
>
> Following your suggestion, we explored a new domain where ***“simulators provide structure and inductive biases unavailable to other methods,”*** and the domain is also more closely aligned with ***“robotics or physical simulation”*** noted in your original review. The results show that Sim-PE performs well in this setting. We hope these new experiments help address your remaining concerns.
>
> ## **Section 1. Domain Description**
>
> We consider a setting where we are given a dataset of *observations* collected from a system (e.g., robot sensor readings, surrounding video from a car, or game screenshots). The goal is to generate a dataset of *control signals* (e.g., robot control commands, steering inputs for a car, or game actions) such that, when these signals are executed in the system, the resulting distribution of observations matches that of the given dataset. The generated control signals must satisfy differential privacy with respect to the observation dataset. We additionally assume access to a simulator that can approximate the behavior of the system, which is common in these domains.
>
> ### **Section 1.1. Why existing techniques cannot be directly applied**
>
> Existing techniques such as DP fine-tuning or DP tabular methods typically assume that the input data and the generated data lie in the same domain (e.g., image→image, text→text, tabular→tabular). In our case, the given data (observations) and the target data (control signals) lie in **different** domains. Although the simulator maps control signals to observations, its non-differentiable nature makes it unclear how to adapt these existing methods to this setting.
>
> In contrast, Sim-PE naturally fits this problem. The optimization space of Sim-PE is the simulator’s parameters—which, in this application, correspond to the control signals. Consequently, Sim-PE can be applied both when the target generated data lies in the simulator’s *output* space (as in the image and networking experiments in the submission and rebuttal) and when it lies in the simulator’s *input* space (as in this new application).
>
> ## **Section 2. Experimental Settings**
>
> ### **Section 2.1. Data and Simulator**
> We use the popular Gymnasium library (https://github.com/Farama-Foundation/Gymnasium, previously known as the Gym library by OpenAI) for our experiments. Gymnasium provides a wide range of simulated environments for robotics, control, physics, and games, originally designed for developing and benchmarking reinforcement learning algorithms. Since these environments align well with the target domain described above, we adapt them for our setting.
>
> We use the **CartPole** environment (https://gymnasium.farama.org/environments/classic_control/cart_pole/) in the Gymnasium library as the example. In this environment, a robot cart balances a pole placed vertically on top of it. At each timestep, the *observation* consists of four quantities: cart position, cart velocity, pole angle, and pole angular velocity. A *control signal* moves the cart left or right to keep the pole balanced. The episode terminates when the pole angle exceeds ±12° or the cart moves outside the display. The task is considered successful if the pole remains balanced for 500 frames; otherwise it fails. To map observations to control signals, we adopt a simple linear policy of the form
> `sign(<observations, policy>)`, where **policy** is a 4-dimensional vector and `<·,·>` denotes the inner product.
>
> To create a “private dataset,” we enumerate all policies with each coordinate in `{−5, −4.5, …, 5}` (a total of \(21^4\) policies). We then uniformly sample 2500 policies that succeed (label = 1) and 2500 policies that fail (label = 0).
>
> ### **Section 2.2. Embedding**
> We use raw observations as the embedding. Since episodes may terminate at different lengths (due to success or failure), we pad all observation sequences to a fixed length of 500 frames by repeating the final observation. To keep it simple, we use the raw values directly, without any normalization, and we find that this simple embedding can already yield strong results. This further answers the reviewer’s original question regarding embedding models: it shows that Sim-PE can still function effectively without any pretrained embeddings.
>
> ### **Section 2.3. RANDOM and VARIATION APIs**
> We follow the same procedure as in Section 3.2:
> * **RANDOM API** samples policy parameters uniformly at random.
> * **VARIATION API** perturbs existing policy parameters.

---

> > ### Author Response · Authors · 2025-11-25
> > **Additional Experiments in Domains where Sim-PE could Offer Unique Advantages (2/2)**
> >
> > ### **Section 2.4. Metrics**
> > * **Utility:** Recall that the private dataset contains two classes (success = 1, failure = 0). For each generated policy, we run it in the simulator to determine whether it succeeds or fails, compare this outcome to the assigned label, and report the *accuracy*. This metric directly aligns with the objective of the environment—successfully balancing the pole.
> >
> > * **Fidelity:** We compute the Wasserstein distance between the distributions of episode lengths (i.e., observation sequence lengths) in the generated and private datasets. This provides a more fine-grained measure than utility alone, as episode lengths can vary significantly even among failed policies.
> >
> > ## **Section 3. Results**
> >
> > We summarize our findings around three research questions:
> > (1) How does synthetic data quality evolve across PE iterations?
> > (2) How does the privacy budget affect performance?
> > (3) How efficient is Sim-PE in this setting?
> >
> > **(1) How does synthetic data quality change across PE iterations?**
> > We first fix the privacy budget to ε = 1. The table below reports the utility and fidelity metrics across all PE iterations. We can see that both metrics steadily improve over time. More specifically: (1) since random policies fail roughly 99% of the time, the initial accuracy is near 50%. Sim-PE progressively discovers policies that match the success/failure patterns of the private data, ultimately reaching 76% accuracy. (2) The Wasserstein distance also decreases by roughly 4× over the iterations. These results demonstrate that Sim-PE performs effectively in this setting.
> >
> > | Iteration | Wasserstein Distance | Accuracy |
> > |-----------|-----------------------|----------|
> > | 0         | 234.72                | 0.50     |
> > | 1         | 207.53                | 0.51     |
> > | 2         | 166.90                | 0.51     |
> > | 3         | 134.13                | 0.54     |
> > | 4         | 121.93                | 0.59     |
> > | 5         | 114.21                | 0.62     |
> > | 6         | 93.75                 | 0.66     |
> > | 7         | 76.12                 | 0.70     |
> > | 8         | 75.78                 | 0.73     |
> > | 9         | 60.50                 | 0.76     |
> >
> >
> > **(2) How does the privacy budget affect performance?**
> > We vary the privacy budget from 0.2 to 8 to evaluate how Sim-PE behaves under different privacy constraints. The results below show a clear trend: performance improves as the privacy budget increases. With ε = 8, the accuracy reaches 96%, a very strong result. This confirms that Sim-PE can leverage larger privacy budgets effectively when they are available.
> >
> > | Epsilon | Wasserstein Distance | Accuracy |
> > |---------|-----------------------|----------|
> > | 0.1     | 151.46                | 0.55     |
> > | 0.2     | 142.75                | 0.65     |
> > | 0.5     | 92.16                 | 0.75     |
> > | 1.0     | 60.50                 | 0.76     |
> > | 2.0     | 37.61                 | 0.86     |
> > | 4.0     | 35.44                 | 0.93     |
> > | 8.0     | 32.52                 | 0.96     |
> >
> > **(3) How efficient is the Sim-PE algorithm?**
> > All experiments above were run entirely on CPU, without any GPU acceleration. On a server equipped with an AMD EPYC 7v12 64-core processor, each full experiment completes in approximately 4 minutes, showing that Sim-PE is highly efficient in this domain.
> >
> > ---
> >
> > In summary, we hope these experiments help illustrate the **unique advantages of Sim-PE** in scenarios where existing approaches cannot be easily applied. **We will remain active through the end of the discussion period.** If you have any additional questions, please feel free to let us know.
> >
> > Thank you again for your thoughtful and constructive suggestions!

---

### Official Review · Reviewer_SEY6 · 2025-10-30

**Soundness:** 3
**Presentation:** 4
**Contribution:** 3
**Rating:** 6
**Confidence:** 3

**Summary:**

This paper addresses a key limitation of Private Evolution (PE), a state-of-the-art framework for generating differentially private (DP) synthetic data. The standard PE framework relies on inference APIs from foundation models (FMs), but its performance degrades significantly when a suitable FM for the private data domain is unavailable (e.g., using an ImageNet-trained FM for MNIST data).

The authors' key insight is that the PE framework is general and only requires two abstract APIs: a `RANDOM_API` to generate initial samples and a `VARIATION_API` to produce variations of a given sample. The paper proposes **Sim-PE**, a novel approach that implements these APIs using non-neural-network data synthesizers, which the authors refer to as "simulators".

The paper introduces two practical methods for integrating simulators:
1.  **Simulator Access:** When the simulator's code is available, `RANDOM_API` renders images with random parameters, and `VARIATION_API` renders images with slightly perturbed parameters.
2.  **Data Access:** When only a large, simulator-generated dataset is available (e.g., due to proprietary assets), `RANDOM_API` samples a random image from the dataset, and `VARIATION_API` selects a random nearest neighbor of a given image.

The authors also demonstrate that simulators and FMs can be used together in a hybrid approach, for instance, by using a simulator for diverse initialization and an FM for refinement.

Experiments show that Sim-PE is highly effective.
- On MNIST, where standard PE fails (27.9% accuracy), Sim-PE with a text-rendering simulator achieves 89.1% accuracy, a 3x improvement.
- On CelebA, Sim-PE successfully selects high-utility samples from a public dataset and a hybrid approach with a weak simulator outperforms using either the simulator or the FM alone.
- Finally, Sim-PE is shown to be up to 80x more computationally efficient than standard PE.

**Strengths:**

The core idea is simple and elegant. While not groundbreaking, this paper opens up PE to a new class of generative tools beyond foundation models.
The presentation is very clear. I especially appreciate the underlying motivation for the choice of experiments as well as the ablation studies.

**Weaknesses:**

- While the authors show that it is not necessary to have a foundation model that is aligned with the private data, it doesn't directly solve the issue of cold starts, but instead shifts the dependency on a good foundation model to a good simulator.

**Questions:**

-  The hybrid model is very promising. Have the authors considered a "mixed" or "parallel" strategy? For example, within a single iteration, could the variation API somehow combine a simulator and FM? This seems possible given the framework's modularity.
- How sensitive is the performance of SIM-PE to the choice of simulator? While ImageNet FM -> private MNIST has low utility due to a mismatch with the foundation model, how does the utility of SIM-PE degrade with mismatched simulators?

---

> ### Author Response · Authors · 2025-11-13
> **Rebuttal (1/2)**
>
> *We sincerely thank the reviewer for recognizing that the paper addresses a key limitation of PE, demonstrates a highly effective method, and introduces a simple and elegant core idea that opens up PE to a new class of generative tools beyond foundation models. We also appreciate the positive feedback on the clarity of the presentation. In addition, thank you very much for your insightful comments. We have conducted all the suggested experiments, which were very helpful. We provide responses to each of your points below.*
>
> ----
>
> ----
>
> > ### Q1. cold starts
>
> First, we would like to clarify the interpretation of *shifts the dependency on a good foundation model to a good simulator*. The goal of Sim-PE is **not** to replace PE with simulators, but to *complement* it. With Sim-PE, PE can now be applied to domains that have either good foundation models **or** good simulators. We believe this expansion of applicability is already a meaningful and sufficient contribution.
>
> Of course, there will be domains where neither a suitable foundation model nor a reliable simulator exists. To address this “cold start’’ scenario, our current direction is to combine DP fine-tuning on a small model (which can adapt more effectively to private data) with Private Evolution. We have preliminary promising results in this (orthorgonal) direction. Due to anonymity constraints, we cannot provide citations during the review period, but we will add a discussion once the review process concludes.
>
> > ### Q2. Have the authors considered a "mixed" or "parallel" strategy?
>
> Thank you very much for sharing this interesting idea!
>
> **Experiment design.**  Following the suggestion, we implemented a mixed strategy where the `RANDOM_API` calls *both* the foundation model (FM) and the simulator. The synthetic samples produced by both are then jointly used for voting. During later iterations, the corresponding `VARIATION_API` (FM or simulator) is called depending on which source generated the synthetic sample.
>
> **Results.**  We conducted experiments on the CelebA dataset using the graphics-based simulator and the same diffusion model as in Section 4.3. We found that in the first iteration (immediately after `RANDOM_API`), **98.6% of the votes from the private data selected simulator-generated samples**. As a result, this mixed strategy would behave almost identically to Sim-PE using only the simulator.
>
> **Analysis.**  This outcome is expected: simulator-generated images (Fig. 3(b)) are much closer to the private data (Fig. 3(a)) than FM-generated images, since FMs are trained on diverse ImageNet categories. However, this *does not* mean that the FM is useless. When given a low-quality image, the FM can improve its visual fidelity through `VARIATION_API`. Our “simulator first, then FM’’ strategy leverages this refinement ability and leads to better performance in practice.
> That said, exploring more sophisticated ways to combine simulators and FMs within each iteration is an interesting direction, and we consider it promising future work. We sincerely thank the reviewer again for this insightful suggestion!

---

> > ### Author Response · Authors · 2025-11-21
> > **Rebuttal (2/2)**
> >
> > > ### Q3. How sensitive is the performance of SIM-PE to the choice of simulator?
> >
> > Thank you for this important question. We already provide detailed experiments on this in the main paper (Sections 4.3 and 4.4). In these experiments, we fix the private dataset to CelebA and evaluate two simulators: a good computer-graphics–based simulator (Fig. 3(b)) and a poor-quality simulator that produces only cartoon-style images (Fig. 4(b)). The key findings are:
> > * **The bad simulator does degrade performance, but is better than some baselines**. From Table 2, we can see that Sim-PE with the bad simulator has worse performance than Sim-PE with the good simulator. However, Sim-PE with this bad simulator still learns meaningful features (e.g., distinguishing long vs. short hair for both women and men in Fig. 4(c)) and achieves better data-quality metrics than several baselines (Tables 2 and 1). This demonstrates the robustness and effectiveness of Sim-PE even under severely suboptimal simulators.
> > * **Even poor simulators can benefit PE through increased diversity.** We study the hybrid use of simulators and foundation models (Secs. 3.4 and 4.4), leveraging the complementary strengths of both: simulators provide diversity while foundation models provide fidelity. Our experiments show that although the bad simulator performs poorly in isolation, it still improves PE’s performance when combined with a foundation model (Table 2).
> >
> > Moreover, in Appendix F.7 of our submission, we include a more fine-grained experiment in which we gradually shift the simulator’s output distribution away from the private data. As expected, as the simulator becomes less aligned with the private data, the quality of Sim-PE’s generated samples also decreases. However, in all settings, Sim-PE consistently achieves lower FID than the simulator’s raw output, demonstrating its ability to identify and select useful synthetic samples even when the simulator is suboptimal.
> >
> >
> > ----
> >
> > ----
> >
> > *Finally, we would like to express our sincere appreciation for your thorough evaluation and insightful suggestions. We are grateful for the time you dedicated to improving our paper.*

---

> > > ### Author Response · Authors · 2025-11-26
> > > **New Experiments**
> > >
> > > **Dear Reviewer SEY6,**
> > >
> > > Thank you again for your thoughtful suggestions and the positive feedback on our work.
> > >
> > > We would like to share an update that, motivated by the important suggestions from **Reviewer kg1H**, we have conducted and added a **new experiment in a challenging application domain where existing DP synthetic data methods cannot be applied** ([link](https://openreview.net/forum?id=SiqK9UzqNT&noteId=2u7o4fpXoK)). This setting highlights a scenario in which **Sim-PE is uniquely effective**, precisely because it can exploit simulator structure in ways that prior DP fine-tuning and tabular-data approaches cannot. We believe this experiment demonstrates **a key advantage and practical impact of Sim-PE**, further reinforcing the novelty and value of our contribution.
> > >
> > > We will remain active throughout the discussion period, and if you have any questions, please feel free to let us know.
> > >
> > > Thank you very much for your time and feedback.
> > >
> > > **The authors**

---

### Official Review · Reviewer_XMoB · 2025-10-31

**Soundness:** 3
**Presentation:** 4
**Contribution:** 4
**Rating:** 6
**Confidence:** 4

**Summary:**

This work extends Private Evolution framework from foundation model APIs to simulators and simulator-generated datasets. Key insight: PE's privacy guarantee depends only on DP Nearest Neighbors Histogram, so generation backend can be abstracted via RANDOM and VARIATION APIs. For accessible simulators, APIs implemented through parameter perturbation. For data-only simulators, nearest-neighbor variation avoids wasting privacy budget on distant samples. On MNIST with ε=1, Sim-PE achieves 89.1% classification accuracy versus PE's 27.9%, reducing FID by 80%. Framework enables combining simulators with foundation models.

**Strengths:**

- Backend-agnostic design demonstrates real impact. MNIST shows foundation model pretrained on ImageNet achieves 27.9% on digits while text renderer achieves 89.1% (Table 1). When distribution mismatch exists, appropriate backend selection matters.
- Handles realistic deployment scenarios: accessible simulator with direct parameter control, and data-only simulator common when proprietary assets involved.
- Nearest-neighbor iterative refinement (Section 3.3) avoids privacy budget waste on irrelevant samples. Table 3 shows clear improvement over naïve approaches.
- Combination strategy (Table 2) demonstrates framework flexibility: simulator + foundation model outperforms either alone on CelebA (FID 11.9 vs 22.0 for PE alone, 99.5 for weak simulator alone).

**Weaknesses:**

- My main concern is with the privacy accounting for the data-based Sim-PE (Sec 3.3). The method uses an external embedding model (Inception, Appendix F.1) for its nearest-neighbor search, but the privacy properties of this embedding are never analyzed. If this embedding was trained on related private data, it could leak information, and this seems to be an unaddressed gap in the privacy proof.

**Questions:**

- Could the authors clarify the privacy accounting for the external embedding model? How can we be sure it doesn't leak information, and shouldn't it be consistent across all baselines for a fair comparison?
- How sensitive is the data-based Sim-PE to the choice of the embedding model? Does performance change significantly if using CLIP, DINOv2, or other task-specific embeddings?

---

> ### Author Response · Authors · 2025-11-13
> **Rebuttal (1/2)**
>
> *We sincerely thank the reviewer for recognizing the method’s real impact, its ability to handle realistic deployment scenarios, and the flexibility of the framework. We also greatly appreciate your insightful comments. We have conducted all the suggested experiments. We provide responses to each of your points below.*
>
> ----
>
> ----
>
> > ### Q1.  Privacy accounting for the external embedding model. Consistent across all baselines for a fair comparison
>
> This is a great question!
> * **First, we would like to clarify that our experimental setup is fair.** The Inception embedding used by Sim-PE is pretrained on ImageNet, and **all baselines in our comparison also treat ImageNet as public data** and rely on it in similar ways:
>     * PE uses the same ImageNet-pretrained Inception embedding.
>     * All other training-based baselines pretrain their models on ImageNet (as reported in DPImageBench: https://arxiv.org/pdf/2503.14681, from which we take the baseline results).
> * **Sim-PE’s assumption is reasonable.** Sim-PE assumes that the data used to train the embedding model is public. This mirrors the common assumption in DP fine-tuning methods, which presume that the data used to pretrain large models is public and non-private. Moreover, because pretraining modern models often requires massive datasets, DP fine-tuning approaches sometimes rely on much larger and potentially sensitive corpora—such as LAION (see Table 1 in https://arxiv.org/pdf/2305.15759). In contrast, Sim-PE depends only on an Inception embedding trained on ImageNet, making its assumption much weaker.
> * **In domains without public data or strong embedding models, Sim-PE may still work.** To demonstrate this, we evaluate Sim-PE on a networking dataset, where no good embedding model exists (see the following section). Instead of using a learned embedding space, we compute distances directly in the raw sample space. This shows that Sim-PE might remain applicable even when public embeddings are unavailable.
>
> > ### New experiments on networking data
>
> We follow the experimental setup of NetDPSyn [1], a recent paper published at IMC 2024, one of the top networking conferences.
>
> **Data.**  We aim to generate differentially private *network flow header* data using the TON dataset [1] as the private dataset. Each sample contains the following fields:
> - Source IP address
> - Destination IP address
> - Source port
> - Destination port
> - Protocol (e.g., UDP, TCP)
> - Timestamp of the first packet
> - Duration of the flow
> - Number of packets
> - Number of bytes
> - Flow label (10 classes, e.g., normal, background, DDoS)
>
> **Simulator.**  We use a very simple simulator: the `RANDOM_API` generates random flows, and the `VARIATION_API` perturbs individual fields of a flow (e.g., modifying the IP address).
>
> **Embedding.**  To keep the setup minimal, we embed samples directly in their normalized raw format (one-hot encodings for categorical fields and normalized numeric values for continuous ones). Thus, **the embedding does not rely on any machine learning model**.
>
> **Metrics.**  Following [1], we report:
> - **1-way and 2-way Total Variation Distance (TVD)** between synthetic and private data (lower is better)
> - **Classification accuracy**, obtained by training a classifier on synthetic data and testing on held-out private data (higher is better). This represents the downstream task of network flow data.
>
> **Privacy budget.**  As in [1], we set ε = 2.
>
> **Results.**  The table below shows how the metrics evolve across Sim-PE iterations. Key observations:
>
> 1. **All metrics steadily improve across iterations.** Final performance is significantly better than the initial random samples (e.g., accuracy improves by **16.76%**). This demonstrates that Sim-PE can work effectively for networking data *even without* ML-based embeddings or a sophisticated simulator.
>
> 2. **Compared to Figure 3 in [1], this simple approach already outperforms two baselines in accuracy.** Notably, NetShare—a widely used DP fine-tuning–based synthesizer in networking (130+ citations)—is surpassed by Sim-PE in classification accuracy. This highlights the strong potential of Sim-PE beyond the image domain.
>
> |Iteration| Accuracy|1-way TVD|2-way TVD|
> |--|--|--|--|
> | 0       | 56.48    | 0.29      | 0.46      |
> | 1       | 56.48    | 0.29      | 0.45      |
> | 2       | 56.48    | 0.28      | 0.44      |
> | 3       | 56.75    | 0.27      | 0.43      |
> | 4       | 57.08    | 0.27      | 0.42      |
> | 5       | 58.28    | 0.26      | 0.41      |
> | 6       | 60.29    | 0.25      | 0.40      |
> | 7       | 61.28    | 0.24      | 0.38      |
> | 8       | 63.47    | 0.23      | 0.37      |
> | 9         | 65.95    | 0.22      | 0.36      |
> | 10        | 67.37    | 0.21      | 0.35      |
> | 11        | 68.47    | 0.21      | 0.33      |
> | 12        | 72.18    | 0.20      | 0.32      |
> | 13        | 72.77    | 0.19      | 0.31      |
> | 14        | 73.24    | 0.18      | 0.30      |
>
> [1] https://arxiv.org/abs/2409.05249

---

> ### Author Response · Authors · 2025-11-13
> **Rebuttal (2/2)**
>
> > ### Q2. How sensitive is the data-based Sim-PE to the choice of the embedding model?
>
> Thank you for the suggestion! Following it, we ran Sim-PE with a CLIP embedding. The comparison between the original PE, Sim-PE with the Inception embedding, and Sim-PE with the CLIP embedding is shown in the table below. We observe that switching from Inception to CLIP does not significantly affect performance, and Sim-PE continues to outperform the original PE by a substantial margin.
>
> | Method              | Accuracy |
> |---------------------|----------|
> | PE                  | 70.5%    |
> | Sim-PE (Inception)  | 80.0%    |
> | Sim-PE (CLIP)       | 77.4%    |
>
> ----
>
> ----
>
> *Finally, we would like to express our sincere appreciation for your thorough evaluation and insightful suggestions. We are grateful for the time you dedicated to improving our paper.*

---

> ### Comment · Reviewer_XMoB · 2025-11-23
>
> Thanks for the detailed rebuttal; I’m satisfied that the algorithmic and fairness concerns around the embedding choice are addressed, and I’m happy to keep my overall positive recommendation. My only remaining request is that the final version clearly state the threat model around the embedding (that it is trained only on public data and independent of the protected dataset), so readers don’t over‑interpret the DP guarantee as covering the embedding pretraining as well.

---

> > ### Author Response · Authors · 2025-11-26
> >
> > **Dear Reviewer XMoB,**
> >
> > We are pleased to hear that your concerns have been addressed. Following your suggestion, we have **updated the revision** and added our assumption regarding the embedding model in **Appendix A** (with changes highlighted in green).
> >
> > We would also like to share that, motivated by the important suggestions from **Reviewer kg1H**, we have conducted a **new experiment in a challenging application domain where existing DP synthetic data methods cannot be applied** ([link](https://openreview.net/forum?id=SiqK9UzqNT&noteId=2u7o4fpXoK)). This setting showcases a scenario in which **Sim-PE is uniquely effective**, precisely because it can leverage simulator structure in ways that prior DP fine-tuning or tabular-data approaches cannot. We believe this experiment highlights **a key advantage and practical impact of Sim-PE**, reinforcing the novelty and value of our contribution.
> >
> > Given these revisions and the addition of this new experiment, we kindly ask whether you might consider increasing your score.
> >
> > Thank you very much for your time and feedback.
> >
> > **The authors**

---

### Official Review · Reviewer_BegZ · 2025-11-01

**Soundness:** 3
**Presentation:** 3
**Contribution:** 2
**Rating:** 2
**Confidence:** 4

**Summary:**

The paper introduces simPE, an extension of PE. In traditional PE, the work relies on foundational models. In simPE, the authors show that PE framework can be extended to other simulators as well and not rely on only foundational models.

**Strengths:**

The paper is well written and easy to understand. The work proposed is simple to use but powerful and extends PE to work with simulators. The work also provides clear, actionable guidelines for both simulator access and data-only scenarios.

**Weaknesses:**

1. The paper's contribution is minimal. It extends the same framework of PE to work with simulators. Since PE itself needs only APIs, the main contribution of the paper seems to be replacing foundational models behind the APIs with simulators.
2. The work focuses mainly on image generation because of dependencies on simulators. The quality of the images also depend on the availability of the simulator's capability to generate data similar to the original dataset.

**Questions:**

1. How sensitive is Sim-PE to simulator quality?
2. How do biases in simulators transfer to the synthetic data? Are there ways to detect or mitigate such transfers?

---

> ### Author Response · Authors · 2025-11-13
> **Rebuttal (1/3)**
>
> *We sincerely thank the reviewer for recognizing that the paper is well written, easy to follow, and that the proposed approach is both simple and powerful. We also greatly appreciate your insightful comments. We have conducted all the suggested experiments, which were very helpful. We provide responses to each of your points below.*
>
> ----
>
> ----
>
> > ### Q1. The paper's contribution is minimal.
>
> **We respectfully but strongly disagree with the reviewer’s assessment of our contribution.** Below, we clarify our perspective on Sim-PE and, more broadly, on the trajectory of DP synthetic data research.
>
> **1. Using simulators for DP synthetic data has never been explored before.**  The central idea—that *PE can operate effectively with simulators*—is both non-obvious and fundamentally new. Among the 60+ follow-up works citing PE (per Google Scholar), *none* investigate non–foundation-model data synthesizers such as simulators. More broadly, throughout the entire DP synthetic data literature, the simulators we explored have never been used as a DP data generator. Sim-PE therefore not only broadens the applicability of PE, but also introduces a new paradigm that unlocks the potential of simulators for DP data generation. We believe this reframes an important part of the design space for future work in DP synthetic data.
>
> **2. Simplicity is a strength, not a weakness, and this seemingly simple approach is non-trivial to devise.**  We strongly disagree with the implication that a “simple approach’’ is inherently non-novel or insufficient for acceptance. As detailed in Section 3.3 and empirically validated in Appendix F.1, the most straightforward strategies fail for this problem; arriving at this “simple” solution required careful reasoning and substantial insight. The fact that such an easy-to-implement idea was missed by all prior work highlights its conceptual novelty rather than detracting from it.
> Moreover, simplicity is a practical strength that makes methods easier to use, deploy, and build upon, rather than a weakness.
>
> **3. Fairness and consistency in evaluating contributions.**  Before Private Evolution (PE), DP fine-tuning was the dominant paradigm for DP synthetic data generation. Most works in this line rely on *the same DP mechanism*, DP-SGD (proposed in 2016), and many famous and important works differentiate themselves primarily by *changing the models*. Representative examples include:
>
>     - *Differentially Private Fine-Tuning of Language Models* (Yu et al., ICLR 2022, >500 citations): DP-SGD on RoBERTa and GPT2.
>     - *Synthetic Text Generation with Differential Privacy: A Simple and Practical Recipe* (Yue et al., ACL 2023 Honorable Mention, >100 citations): DP-SGD on GPT2.
>     - *Differentially Private Diffusion Models* (Tim et al., TMLR 2023, >150 citations): DP-SGD on diffusion models.
>     - PrivImage (Li et al., USENIX Security 2024): DP-SGD on diffusion models.
>     - *Differentially Private Latent Diffusion Models* (Liu et al., TMLR 2024): DP-SGD on latent diffusion models.
>
> Conceptually, PE plays the same role as DP-SGD—a *DP mechanism* for synthetic data generation. Under this analogy, *applying DP-SGD on new models is directly parallel to applying PE on new APIs*. Since applying DP-SGD on new models is widely accepted and regarded as a sufficient contribution, we argue that **our work should be treated fairly in the same way.**
>
> **Summary.**  Across all dimensions—fairness in evaluating contributions relative to prior work, the non-obviousness of the proposed approach, the practical strength of the method’s simplicity, the conceptual novelty of enabling PE to operate with simulators, and broader impact of introducing simulator-based generation into the DP landscape—we believe the contribution of Sim-PE is both solid and significant. We also hope that **our new experiments demonstrating Sim-PE’s applicability in a completely different domain (networking)** help reassure the reviewer of the value and potential of the approach (see Q3).
>
> We thank the reviewer for sharing their perspective and welcome further discussion.

---

> ### Author Response · Authors · 2025-11-13
> **Rebuttal (2/3)**
>
> > ###  Q2. The quality of the images also depend on the availability of the simulator's capability to generate data similar to the original dataset. How sensitive is Sim-PE to simulator quality?
>
> Thank you for this important question. We already provide detailed experiments on this in the main paper (Sections 4.3 and 4.4). In these experiments, we fix the private dataset to CelebA and evaluate two simulators: a good computer-graphics–based simulator (Fig. 3(b)) and a poor-quality simulator that produces only cartoon-style images (Fig. 4(b)). The key findings are:
> * **The bad simulator does degrade performance, but is better than some baselines**. From Table 2, we can see that Sim-PE with the bad simulator has worse performance than Sim-PE with the good simulator. However, Sim-PE with this bad simulator still learns meaningful features (e.g., distinguishing long vs. short hair for both women and men in Fig. 4(c)) and achieves better data-quality metrics than several baselines (Tables 2 and 1). This demonstrates the robustness and effectiveness of Sim-PE even under severely suboptimal simulators.
> * **Even poor simulators can benefit PE through increased diversity.** We study the hybrid use of simulators and foundation models (Secs. 3.4 and 4.4), leveraging the complementary strengths of both: simulators provide diversity while foundation models provide fidelity. Our experiments show that although the bad simulator performs poorly in isolation, it still improves PE’s performance when combined with a foundation model (Table 2).
>
> Moreover, in Appendix F.7 of our submission, we include a more fine-grained experiment in which we gradually shift the simulator’s output distribution away from the private data. As expected, as the simulator becomes less aligned with the private data, the quality of Sim-PE’s generated samples also decreases. However, in all settings, Sim-PE consistently achieves lower FID than the simulator’s raw output, demonstrating its ability to identify and select useful synthetic samples even when the simulator is suboptimal.

---

> ### Author Response · Authors · 2025-11-13
> **Rebuttal (3/3)**
>
> > ### Q3.  The work focuses mainly on image generation because of dependencies on simulators.
>
> Although our current experiments focus on image generation, we would like to emphasize that Sim-PE can apply more broadly across domains, as discussed in Section 5 and Appendix I. To illustrate this, we conduct an additional experiment on **networking data**, following the experimental setup of NetDPSyn [1], a recent paper published at IMC 2024, one of the top networking conferences.
>
> **Data.**  We aim to generate differentially private *network flow header* data using the TON dataset [1] as the private dataset. Each sample contains the following fields:
> - Source IP address
> - Destination IP address
> - Source port
> - Destination port
> - Protocol (e.g., UDP, TCP)
> - Timestamp of the first packet
> - Duration of the flow
> - Number of packets
> - Number of bytes
> - Flow label (10 classes, e.g., normal, background, DDoS)
>
> **Simulator.**  We use a very simple simulator: the `RANDOM_API` generates random flows, and the `VARIATION_API` perturbs individual fields of a flow (e.g., modifying the IP address).
>
> **Embedding.**  To keep the setup minimal, we embed samples directly in their normalized raw format (one-hot encodings for categorical fields and normalized numeric values for continuous ones). Thus, **the embedding does not rely on any machine learning model**.
>
> **Metrics.**  Following [1], we report:
> - **1-way and 2-way Total Variation Distance (TVD)** between synthetic and private data (lower is better)
> - **Classification accuracy**, obtained by training a classifier on synthetic data and testing on held-out private data (higher is better). This represents the downstream task of network flow data.
>
> **Privacy budget.**  As in [1], we set ε = 2.
>
> **Results.**  The table below shows how the metrics evolve across Sim-PE iterations. Key observations:
>
> 1. **All metrics steadily improve across iterations.** Final performance is significantly better than the initial random samples (e.g., accuracy improves by **16.76%**). This demonstrates that Sim-PE can work effectively for networking data *even without* ML-based embeddings or a sophisticated simulator.
>
> 2. **Compared to Figure 3 in [1], this simple approach already outperforms two baselines in accuracy.** Notably, NetShare—a widely used DP fine-tuning–based synthesizer in networking (130+ citations)—is surpassed by Sim-PE in classification accuracy. This highlights the strong potential of Sim-PE beyond the image domain.
>
> | Iteration | Accuracy | 1-way TVD | 2-way TVD |
> |-----------|----------|-----------|-----------|
> | 0         | 56.48    | 0.29      | 0.46      |
> | 1         | 56.48    | 0.29      | 0.45      |
> | 2         | 56.48    | 0.28      | 0.44      |
> | 3         | 56.75    | 0.27      | 0.43      |
> | 4         | 57.08    | 0.27      | 0.42      |
> | 5         | 58.28    | 0.26      | 0.41      |
> | 6         | 60.29    | 0.25      | 0.40      |
> | 7         | 61.28    | 0.24      | 0.38      |
> | 8         | 63.47    | 0.23      | 0.37      |
> | 9         | 65.95    | 0.22      | 0.36      |
> | 10        | 67.37    | 0.21      | 0.35      |
> | 11        | 68.47    | 0.21      | 0.33      |
> | 12        | 72.18    | 0.20      | 0.32      |
> | 13        | 72.77    | 0.19      | 0.31      |
> | 14        | 73.24    | 0.18      | 0.30      |
>
> [1] https://arxiv.org/abs/2409.05249
>
> > ### Q4. How do biases in simulators transfer to the synthetic data? Are there ways to detect or mitigate such transfers?
>
> Thank you for the great question. Our experiments in Section 4.4 provide a clear example of how biases in simulators can transfer to the synthetic data. When using a simulator with an extremely biased style (Fig. 4(b)), Sim-PE indeed inherits this bias: the generated images reflect the same cartoon-like appearance (Fig. 4(c)). This bias is also captured by our quantitative metrics—both FID and classification accuracy reflect these degraded or skewed results (Table 2).
>
> To mitigate such bias, while still benefiting from the simulator’s contribution to sample diversity, Sections 3.4 and 4.4 explore a **hybrid strategy**: use the simulator only in the early iterations to inject diversity, and then switch to foundation models in later iterations to correct the style and improve image quality. Table 2 shows that this hybrid approach yields strictly better results than using either simulators or foundation models alone, suggesting it is an effective way to reduce simulator-induced bias.
>
> ----
>
> ----
>
> *Finally, we would like to express our sincere appreciation for your thorough evaluation and insightful suggestions. We are grateful for the time you dedicated to improving our paper.*

---

> > ### Author Response · Authors · 2025-11-26
> > **Additional Experiments in Domains where Sim-PE could Offer Unique Advantages**
> >
> > **Dear Reviewer BegZ,**
> >
> > We understand that your primary concern was about the contribution of the work. In addition to the experiments presented in our rebuttal, we would also like to share that—motivated by the important suggestions from **Reviewer kg1H**—we have conducted a **new experiment in a challenging application domain where existing DP synthetic data methods cannot be applied** ([link](https://openreview.net/forum?id=SiqK9UzqNT&noteId=2u7o4fpXoK)). This setting highlights a scenario in which **Sim-PE is uniquely effective**, precisely because it can leverage simulator structure in ways that prior DP fine-tuning or tabular-data approaches cannot. We believe this experiment demonstrates **a key advantage and practical impact of Sim-PE**, further reinforcing the novelty and value of our contribution.
> >
> > We hope this new experiment helps address your remaining concerns. We will remain active through the end of the discussion period, and if you have any additional questions, please feel free to let us know.
> >
> > Thank you very much for your time and feedback.
> >
> > **The authors**

---

### Author Response · Authors · 2025-12-03
**Summary of the Paper and the Discussion**

**Dear Reviewers, ACs, SACs, and PCs,**

We sincerely thank you for overseeing our submission and for the thoughtful and constructive feedback. Below, we first provide a brief summary of the paper, followed by how we addressed the reviewers’ concerns, organized by both questions and reviewers.


## **Paper Summary**

Private Evolution (PE) is a recent training-free framework for DP data synthesis. Prior work focused exclusively on PE with *foundation models*. In this paper, we introduce Sim-PE, which applies PE to *simulators*, such as graphics engines for image synthesis and physics-based simulators for robotics.

**Key contributions:**
* **Novel idea.** We show, for the first time, that PE can operate effectively with simulators—an insight that is non-obvious and required careful method design (Section 3.3).
* **Addressing a core limitation of PE.** Existing PE methods fail in domains without foundation models. Sim-PE expands PE’s applicability to domains where simulators exist.
* **Enabling new DP applications.** During rebuttal, we demonstrated Sim-PE on new applications where **no existing DP synthesis approach can be directly applied**, such as synthetic data with physical simulation (thanks to Reviewer kg1H).

We appreciate that reviewers also recognized these strengths:

1. **Simple and effective approach**
   * **Reviewer BegZ**: “The work proposed is simple to use but powerful”
   * **Review XMoB**: “Backend-agnostic design demonstrates real impact”, “Handles realistic deployment scenarios”, “Combination strategy demonstrates framework flexibility”
   * **Reviewer SEY6**: “The core idea is simple and elegant”
   * **Reviewer kg1H**: “A simple but effective approach”

2. **Good writing**
   * **Review BegZ**: “The paper is well written and easy to understand”
   * **Reviewer SEY6**: “The presentation is very clear”

3. **Insights, results, and impact**
   * **Reviewer BegZ**: “clear, actionable guidelines for both simulator access and data-only scenarios”
   * **Reviewer XMoB**: “clear improvement over naïve approaches”
   * **Reviewer SEY6**: “Sim-PE is highly effective”, “this paper opens up PE to a new class of generative tools beyond foundation models”
   * **Reviewer kg1H**: “Comprehensive experiments”

## **How We Addressed Reviewers’ Concerns (by question)**

1. **Contribution and Novelty (Reviewers BegZ and kg1H)**
We had an in-depth discussion with **Reviewer kg1H** (thank you again), and we **[reached agreement](https://openreview.net/forum?id=SiqK9UzqNT&noteId=CoXDArXE1B)** on adding a new experiment in a domain where **existing DP synthetic data methods do not apply**.
Following this guidance, we added a [new experiment](https://openreview.net/forum?id=SiqK9UzqNT&noteId=2u7o4fpXoK) where the goal is to generate **DP control signals** such that executing them in a system reproduces the distribution of a private observation dataset. Sim-PE naturally supports this task by integrating simulators, whereas existing methods cannot be applied due to the mismatched input/output domains and the non-differentiability of simulators. Importantly, **this experiment uses exactly the same Sim-PE algorithm as in the original submission**; the only difference is the application domain.
We believe this experiment provides a clear and sufficient demonstration of the paper’s contribution, showcasing a setting where **Sim-PE offers unique advantages over existing methods**.

2. **Sensitivity to simulator quality/bias (Reviewers BegZ and SEY6)**
Already included in Sections 4.3, 4.4, and Appendix F.7.

3. **Beyond image data (Reviewers BegZ and kg1H)**
During rebuttal, we added experiments on **[networking data](https://openreview.net/forum?id=SiqK9UzqNT&noteId=8ZV5Qng57K)** and **[physics-based simulation](https://openreview.net/forum?id=SiqK9UzqNT&noteId=2u7o4fpXoK)**

4. **Reliance on embedding models (Reviewers XMoB and kg1H)**
The newly added experiments above **do not use any embedding model**, showing Sim-PE can operate without them.
We further added a [sensitivity analysis](https://openreview.net/forum?id=SiqK9UzqNT&noteId=PpJC7cI715) on the embedding model.

## **How We Addressed Reviewers’ Concerns (by reviewer)**

* **Reviewer XMoB**
Confirmed that all concerns were addressed and maintained their positive recommendation ([link](https://openreview.net/forum?id=SiqK9UzqNT&noteId=fVOepDY6Y2)).

* **Reviewer kg1H**
Their only remaining request was additional experiments in new domains where Sim-PE offers unique advantages. As discussed before, we added the experiments, though the early end of the discussion period prevented further feedback from the reviewer.

* **Reviewers SEY6 and BegZ**
We did not receive responses, but our rebuttal includes point-by-point clarifications to all of their concerns.

---

We sincerely thank the reviewers again for the valuable feedback, which has strengthened the paper, and the ACs and SACs for their careful oversight of the process.

*Authors*

---

### Meta-Review · Area_Chair_Lehf · 2026-01-06

**Summary:**

In this work, the authors discovered that the Private Evolution (PE) framework is sufficiently general to allow APIs beyond foundation models. They found that Prior work focused exclusively on PE with foundation models, and introduced Sim-PE that applies PE to simulators, such as graphics engines for image synthesis and physics-based simulators for robotics.
The reviewers generally recognized that this work is simple and effective.
However, the main concerns include limited novelty and contribution that were posed by Reviewers BegZ and kg1H.
In particular, Reviewer kg1H's concern remains: The PE framework remains essentially unchanged, and adapting its RANDOM and VARIATION APIs to simulators is straightforward. Key limitations of PE persist, including but not limited to mode collapse, sampling bias (e.g., top-K), and non-monotonic performance w.r.t. iterations.  Reviewer kg1H also argues that a submission must be evaluated on its own technical and empirical contributions, not on comparisons to prior publication standards or claims about fairness.
Although the authors indeed spent a lot of efforts to address the reviewers' concerns, it is unfortunately that they cannot convince all the reviewers.
By considering all the submissions that this AC is dealing with, this work cannot be accepted at its current status.

**Reviewer Concerns:**

Reviewer kg1H's concerns were not entirely addressed.

**Reviewer Scores:**

none

---

### Decision · Program_Chairs · 2026-01-26

Reject